# GRIC: General Representation and Informative Content for Enhanced Out-of-Distribution Detection

## Abstract

Out-of-distribution (OOD) detection is crucial for ensuring the robustness of machine learning models in open-world scenarios by identifying inputs from unknown classes. Vision-language models like CLIP have enabled zero-shot OOD detection without requiring labels or training on in-distribution (ID) data. However, current approaches are limited by their dependence on *closed-set text-based labels* and *full image feature representations*, constraining CLIP's capacity to generalize across diverse labels. In this work, we propose GRIC, a novel method based on few-shot multi-modal OOD detection by leveraging two key insights: (1) OOD detection is driven by general ID representations rather than class-specific features, and (2) large language models (LLMs) can enrich the model's understanding of ID data and simulate potential OOD scenarios without actual OOD samples. GRIC demonstrates simplicity and effectiveness, achieving up to a 19% reduction in FPR95 on the MS-COCO dataset and up to 5% on ImageNet, outperforming state-of-the-art methods.

## 1 Introduction

Out-of-distribution (OOD) detection is essential in real-world machine learning applications, where novel, previously unseen classes require special attention. While there has been growing interest in OOD detection, most existing methods rely heavily on single-modal learning approaches (Hendrycks et al., 2020; Hsu et al., 2020; Jin et al., 2022; Shen et al., 2021; Xu et al., 2021; Zhan et al., 2021; Zheng et al., 2020; Zhou et al., 2021) (see Fig. 1(a)). Dependence on visual data alone can be limiting, especially when OOD inputs closely resemble in-distribution (ID) data visually but differ in their semantic content. For example, in image classification tasks, labels are often encoded as one-hot vectors, which fail to capture the rich semantic information embedded in textual descriptions (Ming et al., 2022).

Conventional single-modality OOD detection techniques in vision rely on robust feature representations (Sehwag et al., 2021; Tack et al., 2020) and predefined distance metrics (Lee et al., 2018; Sun et al., 2022) to separate OOD data from ID data. Recent advancements in multi-modal (vision-language) pre-training, such as CLIP (Radford et al., 2021) and ALIGN (Jia et al., 2021), have opened new avenues for learning representations by aligning images with textual descriptions. As a result, OOD detection has begun shifting from traditional single-modal approaches toward multi-modal frameworks (Fort et al., 2021; Esmaeilpour et al., 2022; Ming et al., 2022; Wang et al., 2023). The transition to multi-modal approaches introduces two main challenges:

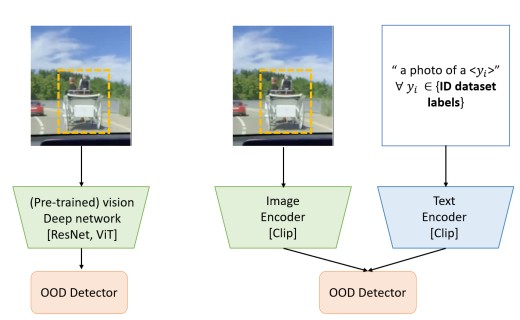

a) Single-Modal (vision) OOD Detection    b) Multi-Modal (vision-language) OOD Detection

**Figure 1:** Different paradigms of OOD detection methods. In multi-modal OOD detection, OODness is defined w.r.t. the task (ID dataset labels).

1) **Defining OODness in a multi-modal framework**, especially when using vision-language models like CLIP, where training datasets are not explicitly disclosed. In such cases, OODness is typically determined based on the task and *textual class names* of specific datasets (e.g., ImageNet class names as ID) (Fort et al., 2021; Esmaeilpour et al., 2022; Ming et al., 2022; Wang et al., 2023).

2) **Identifying specific vision feature segments that integrate well with text features** is critical for multi-modal OOD detection. Methods that add OOD labels into CLIP (Fort et al., 2021) or generate labels using CLIP's visual encoder (Esmaeilpour et al., 2022) focus on predefined OOD labels and small datasets. MCM (Ming et al., 2022) operates without prior OOD knowledge, handling a wide range of real-world datasets. CLIPN (Wang et al., 2023) improves CLIP's ability to differentiate OOD samples but requires computationally intensive training. These approaches primarily rely on full vision feature representations, overlooking the potential benefits of combining *specific* vision feature segments with textual data.

In response to these challenges, we introduce **GRIC**, a post-hoc OOD detection method inspired by low-rank factorization in deep neural networks (Gao et al., 2020; Elhamifar & Vidal, 2013; Li et al., 2019b) and advancements in LLMs and prompt engineering (Novack et al., 2023; Ge et al., 2023). Unlike prior methods that focus on entire vision features, GRIC emphasizes the significance of **general/common features** in representing datasets and distinguishing between ID and OOD data.

Based on key findings from classification and fine-grained classification research (Fei-Fei et al., 2004; Russakovsky et al., 2015; He et al., 2016; Simonyan & Zisserman, 2014), we observe that class-specific features—those critical for recognizing ID data—are often contained within the subspace of the ID data itself. By focusing on features outside of this subspace, which represent a more generalized depiction of ID data, we can better differentiate OOD from ID samples.

GRIC distinguishes OOD samples by leveraging the general/common feature representation of ID data, while masking class-specific features, and leveraging LLMs to access more informative prompts. It incorporates a new scoring function that disregards low-rank subspaces in ID data and enriches prompts with ID super-class information, improving their effectiveness.

Our key contributions include:

- **GRIC: A novel, training-free OOD detection method** that highlights the importance of general ID data representation and informative prompts. GRIC demonstrates consistent performance, operates without the need for downstream fine-tuning, and shows strong generalizability across various tasks.

- **Performance improvement:** GRIC reduces FPR95 by approximately $19\%$ compared to robust baseline methods, demonstrating its superior performance in distinguishing OOD samples while maintaining accuracy for ID data.

- **Comprehensive analysis:** Extensive ablation studies provide insights into the effectiveness and mechanisms of GRIC, offering a deeper understanding of its design and performance.

## 2 BACKGROUND AND NOTATIONS

**Contrastive Vision-Language Pre-Training** In the realm of visual representation learning, models emphasizing vision-language representation demonstrate superior efficacy in image classification tasks compared to those focused solely on visual features, such as ViT (Dosovitskiy et al., 2021). A prominent example is CLIP (Radford et al., 2021), which employs a self-supervised contrastive objective, akin to the InfoNCE loss (Van den Oord et al., 2018). CLIP aligns images with their textual descriptions, utilizing a dual-stream model with a prompt ($p$) text encoder $\mathcal{T} : p \rightarrow \mathbb{R}^d$ (e.g., Transformer (Vaswani et al., 2017)) and an image encoder $\mathcal{I} : x \rightarrow \mathbb{R}^d$ (e.g., ViT (Dosovitskiy et al., 2021)). Despite its success, models like CLIP typically operate in a 'closed-world' setting, conducting zero-shot classification within a predetermined set of classes, even if the input is irrelevant.

**Few-shot and Zero-Shot OOD Detection** In employing a pre-trained model, we establish a classification task with a defined set of class labels, referred to as $\mathcal{Y}_{in}$, which represent the known (ID) classes. These ID classes are defined in relation to the specific classification task, *rather than the classes utilized during pre-training*. Consequently, out-of-distribution (OOD) instances are identified with respect to the ID classes, not the data distribution during pre-training. zero-shot multi-

modal OOD detection (Fort et al., 2021; Ming et al., 2022; Esmaeilpour et al., 2022; Wang et al., 2023) addresses two primary objectives: (1) identifying samples not corresponding to any known (ID) classes, and (2) assigning test samples to known classes when applicable. The OOD detector, denoted as $G(x; \mathcal{Y}in, \mathcal{T}, \mathcal{I}) : \mathcal{X} \rightarrow \{\text{ID}, \text{OOD}\}$, serves as a protective mechanism for the classification model. Our approach (Fig. 1 (b)) leverages a pre-trained model and avoids training on ID samples; however, it is considered a few-shot multi-modal method as it requires access to a limited number of ID samples to compute PCA, without requiring access to their labels. This reliance on minimal ID data allows our method to effectively identify OOD samples while maintaining scalability and efficiency.

**Principal Component Analysis (PCA)** Wold et al. (1987); Joliffe & Morgan (1992); Bro & Smilde (2014) stands as a foundational technique integral to extracting essential information and discerning the low-rank subspace within data. Operating as a potent feature extraction method, PCA transforms original features into linearly uncorrelated variables, known as principal components. These components are meticulously chosen to maximize variance in the dataset, capturing the most informative and discriminative features. In the realm of class-specific classification tasks, PCA plays a pivotal role by identifying principal components contributing significantly to the variance within each class. This focused representation enhances the efficiency of class-specific models, enabling a more streamlined exploration of the intrinsic structure of the data. However, for robust OOD detection, a nuanced shift in focus from class-specific to *general dataset representation* is imperative. Leveraging PCA, we strategically neutralize features associated with specific classes, obtaining the general feature representation essential for effective OOD detection. This approach ensures a more efficient exploration of the intrinsic structure of the data, underscoring its relevance in tasks that demand distinct features for precise classification.

## 3 OUR METHOD: GRIC

Our method, depicted in Fig. 3, establishes a post-hoc OOD detector $G(\cdot)$ by incorporating both the general feature representation of ID data and informative prompts. For a given task with an ID label set $\mathcal{Y}_{\text{in}} = \{y_1, y_2, ..., y_m\}$, we generate a set of text feature vectors $\mathcal{T}(p_i), i \in \{1, 2, 3, .., m\}$, where $p_i$ corresponds to the text prompt "`a photo of a` $\langle y_i \rangle$" for the label (class name) $y_i$.

Our OOD detection method comprises three primary modules, each elucidated in the subsequent sections: 1) general image feature representation, 2) informative text prompts, and 3) the OOD score function.

### 3.1 GENERAL ID IMAGES FEATURES REPRESENTATION

The accuracy of OOD detection is notably enhanced by both vision and text feature representations. To achieve the most informative **vision** feature representation of ID data, we **only** leverage features corresponding to the *general representation of ID data* (GRID) and mask class-specific features. Class-specific features, which reside on boundaries between different classes, are crucial for tasks like image classification and masking them results in more generalization. The general feature representation captures patterns that are shared across all ID classes. These patterns are more general and robust, making them more effective at distinguishing between OOD and ID data.

This approach aligns with principles of data augmentation and regularization, aiming to enhance the model's robustness to variations in the input data that it has not explicitly encountered during training (Dietterich, 1998; Ma et al., 2018). Feature masking serves as a form of regularization, preventing the model from relying excessively on class-specific features.

**Insight:** The intuition behind the general ID representation resonates in a practical scenario akin to Tesla's autopilot encountering a carriage mistaken for a truck (Fig. 2).

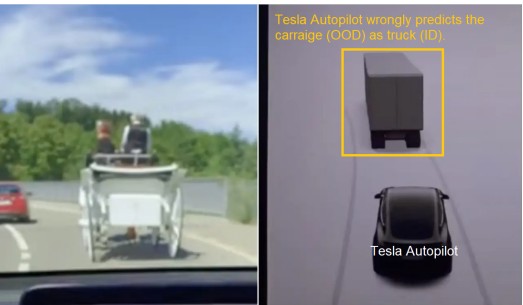

Figure 2: An instance where Tesla Autopilot misidentifies a carriage (OOD) as a truck (ID) underscores the impact of concentrated class-specific features over the general representation of ID data.

Consider the autopilot, trained to discern various vehicle types like trucks, cars, and bicycles (ID). Throughout its training, the model grasps

distinctive features associated with each class. It notably recognizes attributes like wheels and back-side shape.

Now envision the autopilot encountering a carriage (OOD) during its testing and it classifies it as a truck (ID) due to its appearance of specific features like wheels (Fig. 2). To mitigate such misclassifications, our proposed technique involves nullifying the class-specific features, nudging the autopilot to prioritize the broader indicators of a vehicle, encompassing overall size/shape and the presence of a human.

### 3.1.1 EXTRACTION OF GENERAL REPRESENTATION OF ID DATA

As detailed in Section 3, our approach computes the most class-specific feature indices of ID data leveraging Algorithm 1 and masks them in the test phase, assigning them a value of zero to emphasize the general representation of ID data (GRID).

To identify these class-specific features, we first choose $s$ random ID samples from ID dataset and create a representation matrix $R_{ID}^{s \times r}$. It consolidates $s$ representations (each of length $r$) obtained from the forward pass of the ID data through the CLIP image encoder. In the following step, we apply PCA as described in Algorithm 1, yielding principal components (PCA-Components) along with their associated variances. Generally, higher variance corresponds to greater importance in PCA, as components with higher variance explain more of the overall variance in the dataset.

---

**Algorithm 1** Computing the indices of the most important features

1: **function** Compute-Indices(ID data)
2: pca ← PCA()
3: $R_{ID}$ ← CLIP Image Encoder(ID data)
4: pca-components ← pca.fit ($R_{ID}$)
   # Aggregate importance (variance) across all principal components.
5: mean-variance ← Mean(abs(pca-components), axis=0)
   # Sort the features based on their importance.
6: sorted-mean-variance ← Sort(mean-variance, descending)
7: sorted-indices ← argsort(mean-variance, descending)
8: k ← diminish-variance(sorted-mean-variance, threshold)
   # Get k important indices.
9: KI ← sorted-indices[:k]
10: **return** KI
11: **end function**

---

Please refer to Section F of appendix for more details regarding PCA. The PCA-Components form a matrix with dimensions $n_{\text{components}} \times n_{\text{features}}$, where $n_{\text{components}}$ represents the number of principal components, and $n_{\text{features}}$ denotes the total number of features ($r$). To assess the importance of features, we calculate the mean variance of features across the principal components. This computation yields a vector known as the mean-variance (mv) vector (line 5 of Algorithm 1):

$$\text{mean-variance (mv)} = \frac{1}{r} \sum_{i=1}^{r} \left| \text{PCA}(R_i^{s \times r}) \right|. \tag{1}$$

For more details regarding mean variance computation, please refer to Appendix, section G.4.

Subsequently, we organize this vector in descending order and leveraging a criterion for selecting the $k$ most important features. Our criterion involves identifying $k$ such that the rate of changes in mean variance becomes smooth, and the gradient of mean variance approaches zero, which is called mean variance diminishing gradient point. That is, $|\Delta(mv)_k| \leq \epsilon_{th}$, the difference in mean variance between $k$ and $k+1$ features is less than a threshold $\epsilon_{th}$. This convergence criterion recognizes that each additional feature has less variance than the preceding one, emphasizing the importance of the initial ones.

$$SI = \text{argsort}(mv, \text{descending}) \tag{2}$$

$$k = \min\{k' \in SI \mid |\text{mv}[k'] - \text{mv}[k'+1]| \leq \epsilon_{\text{th}} \} \tag{3}$$

The determined value of $k$ and associated principal components pinpoint the indices of class-specific features in the ID representation matrix (Wold et al., 1987; Jolliffe, 2002). We store these $k$ indices in a vector called *KI*, please refer to Fig. 3 (a) and Algorithm 1, line 9. We use KI to mask the corresponding features of test samples for OOD detection. The remaining indices ($r - k$) represent general features across all ID data that play an important role in our method.

$$\text{General representation:} \quad R_i[:, KI] = 0 \quad \text{where} \quad KI = SI[: k]. \tag{4}$$

We apply variance-normalized scaling on general features.

In our OOD detection approach, we present this process by GRID$(x)$. Given a test image $x$, we pass it through the CLIP image encoder and mask features corresponding to the identified indices in $KI$,

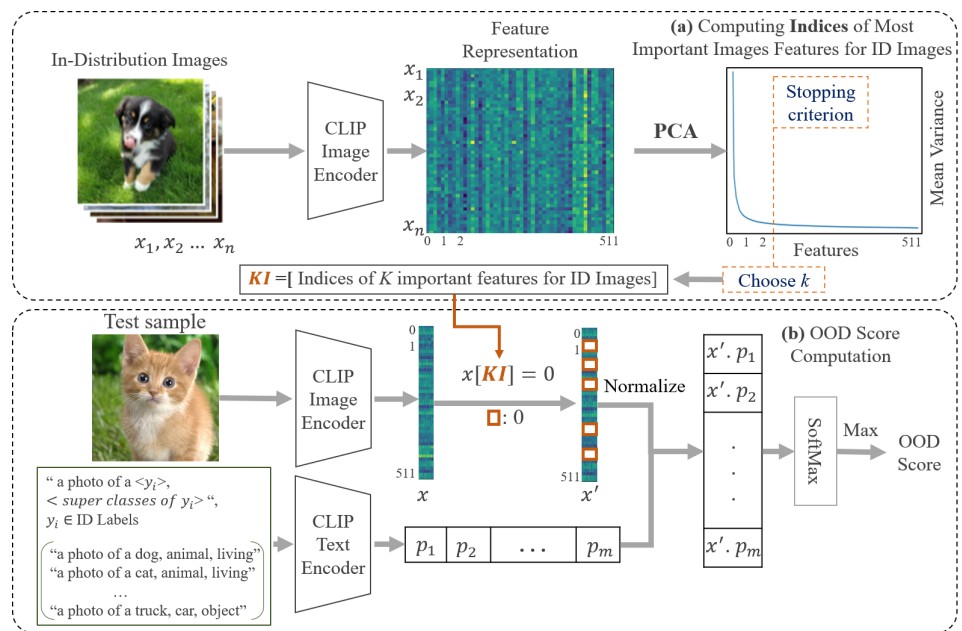

Figure 3: This schematic provides an overview of the proposed training-free multi-modal OOD detection framework. The ID classification task relies on a series of class labels, and the primary objective of OOD detection is to discern samples lying outside this label set. The OOD score for an input image is determined by measuring the distance from the **general visual features** to the nearest **informative contents (prompts) within the ID set**. Careful adjustment of this distance ensures robust separability between ID and OOD instances, as quantified by the GRIC OOD score. For in-depth explanations, please refer to Section 3.

resulting in $x' = \text{GRID}(x)$. Please refer to Fig. 3 (b). This way, we retain the general features while discarding the majority of class-specific features, significantly enhancing the effectiveness of OOD detection. Further details regarding our PCA analysis are provided in the appendix (Section F).

The act of masking the most significant feature representations of ID data with this vector results in the derivation of the general ID representation. In essence, this procedure involves neutralizing features that are highly specific to ID data, allowing for the retention of only those features that contribute to the general representation of ID data.

In our methodology, we purposefully eliminate class-specific features by assigning them a value of zero. This strategic decision aims to mitigate their influence on our OOD score computation. However, the task of masking these features with alternative values poses challenges, as they intricately interact with network weights, potentially leading to varied outcomes, either positive or negative. We explored the utilization of different scaled mean values of features as masks and found no discernible improvement in outcomes.

## 3.2 INFORMATIVE CONTENT/PROMPT (IC)

The efficacy of Contrastive Language-Image Pre-training (CLIP) critically relies on careful prompt engineering, serving as pivotal cues directing the model's comprehension of the intricate interplay between textual descriptions and corresponding visual content (Radford et al., 2021). Thoughtful consideration and formulation of prompts wield considerable influence over the model's capacity to capture nuanced semantics and contextual information (Carion et al., 2020). By tailoring CLIP to specific tasks through prompt engineering, researchers amplify its adaptability and performance across a diverse array of applications, unlocking its full potential in real-world scenarios. The paramount importance of prompt-related considerations becomes evident as practitioners seek optimal results and extend CLIP's applicability in complex, dynamic environments (White et al., 2023; Liu & Chilton, 2022; Zhou et al., 2022c).

In recent investigations into zero-shot multi-modal OOD detection, scant attention has been directed towards the significance of text prompts, as noted by (Fort et al., 2021; Ming et al., 2022; Wang et al., 2023; Shu et al., 2023). Ming et al. (2022) introduces a set of prompts structured as "a photo of a $\langle y_i \rangle$" based on an in-distribution (ID) dataset with $n$ labels. However, this approach simplifies prompts to a class name without incorporating any additional knowledge. Fort et al. (2021) and Esmaeilpour et al. (2022) extend the label set, broadening the exploration scope. However, consid-

ering potential OOD classes or addressing computational hurdles to train an extra text encoder poses challenges. Wang et al. (2023) equip CLIP with the ability to distinguish OOD and ID samples using positive-semantic and negation-semantic prompts, introducing computational costs in training a learnable "no" prompt and a "no" text encoder. However, all these approaches rely only on simple prompt with no extra information.

### 3.2.1 INFORMATIVE PROMPT AUGMENTATION

In our pursuit to improve OOD detection performance, we investigated the augmentation of prompts with supplementary information. A significant challenge arises from the absence of knowledge concerning potential OOD labels. To overcome this hurdle, we utilize existing knowledge, specifically ID labels, and enhance it by incorporating high-level information. This involves identifying the hierarchy within our ID dataset and integrating super-class names into the prompts. For instance, if the ID dataset focuses on dog images, we add "animal" as the super-class name, resulting in prompts like "an image of a dog, animal, living entity.". In our formulation, we present this informative content by IC($y_i$). For this sake, we leverage large language models (LLMs) to generate the taxonomy of ID dataset labels. This simple yet effective approach requires no training or knowledge of OOD data, making it cost-free, straightforward, and effective.

**Insight:** Relying solely on ID labels for OOD classifications presents several limitations. Firstly, it restricts the model's understanding of potential labels and hinders its ability to encompass diverse data points due to the inherent constraint of a limited set of ID labels. Secondly, insufficient exposure to ID general patterns diminishes the model's capacity for comprehensive generalization, making it challenging to recognize and classify novel instances accurately. This narrow scope results in a diminished gap between ID scores and OOD scores, complicating the discernment between in-distribution and out-of-distribution instances.

To overcome these limitations, integrating hierarchical information, such as super-class names, into OOD detection offers significant benefits. It enriches the understanding of ID labels and patterns, facilitating improved generalization capabilities and better recognition of novel instances. Additionally, the incorporation of hierarchical information widens the contextual understanding, enabling a more effective differentiation between in-distribution and out-of-distribution instances. This broader context fosters a stronger association between textual prompts and visual content, enhancing the model's interpretative abilities.

In summary, leveraging hierarchical information in OOD detection enhances knowledge of ID labels and general patterns, improves generalization, widens the context for classification, and enriches the semantic understanding of input data. This integration fosters a robust association between textual and visual information, ultimately enhancing the accuracy and reliability of OOD detection methodologies.

### 3.3 OOD SCORE FUNCTION

Our OOD scoring function involves computing the softmax over the cosine similarity score between general image features (GRID) and informative content (IC). For any test input image $x$, the label-wise matching score, denoted as $s_i(x)$, based on the cosine similarity between the general ID representation of image $\mathcal{GRID}(x)$ and the text features of informative content $\mathcal{IC}(y_i)$ is calculated as follows:

$$s_i(x) = \frac{\mathcal{GRID}(x) \cdot \mathcal{IC}(y_i)}{\|\mathcal{GRID}(x)\| \cdot \|\mathcal{IC}(y_i)\|}. \tag{5}$$

The formal definition of the matching score $S(x; \mathcal{Y}\text{in}, \mathcal{T}, \mathcal{I})$ is given in equation 6. This score effectively determines the match between the input image and the text feature vectors. For ID data, the image is matched to one of the text feature vectors with a high score.

$$S(\mathbf{x}) = \max_i \frac{e^{s_i(\mathbf{x})}}{\sum_{j=1}^N e^{s_j(\mathbf{x})}} \tag{6} \qquad\qquad G(\mathbf{x}; \mathcal{Y}_{\text{in}}) = \begin{cases} 1 & S(\mathbf{x}) \geq \lambda \\ 0 & S(\mathbf{x}) < \lambda \end{cases} \tag{7}$$

The OOD detection function is formally expressed in equation 7. In this formula, conventionally, 1 represents the positive class (ID), and 0 indicates OOD. The threshold $\lambda$ is selected to ensure that a high fraction of ID data (e.g., $95\%$) is above the threshold.

For samples classified as ID, the class prediction can be obtained based on the closest concept: $\hat{y} = \arg\max_{i \in [m]} s_i$. For brevity, we use $S(x)$ to refer to $S(x; \mathcal{Y}_{\text{in}}, \mathcal{T}, \mathcal{I})$ throughout this paper.

As expressed in Equations 5, 6, and 7, the computation of both the similarity score and the OOD score function relies on two essential components: 1) **the *general* representation of image features**, and 2) **the representation of *informative* text prompt features**.

Our approach prioritizes these critical components to enhance the efficacy of the OOD scoring function. This is achieved through a dual emphasis on: a) a representation of general features within ID data, and b) the integration of informative prompts. Incorporating these factors into the similarity and OOD score computations has yielded promising performance.

## 4 EXPERIMENTS

This section provides a comprehensive evaluation of the performance of our method, GRIC, through extensive experiments involving diverse ID and OOD datasets. Following the experimental settings of multi-modal OOD baselines, we elucidate the details of our experimental setup in the subsequent discussion. The empirical studies conducted herein showcase the superior performance of GRIC when compared to existing state-of-the-art baselines, as detailed in Section 4.2. We also present an ablation study in Section 4.3 to provide a more nuanced understanding of our methodology.

### 4.1 DATASETS AND IMPLEMENTATION DETAILS

**Datasets:** In this study, we extend the scope of our evaluations along three key dimensions: (1) image resolution, (2) dataset diversity, and (3) number of classes.

Our selected ID datasets include CUB-200(Wah et al., 2011), FOOD-101(Bossard et al., 2014), OXFORD-PET(Parkhi et al., 2012), and variations of IMAGENET(Deng et al., 2009). For OOD test datasets, we utilize subsets from(Huang & Li, 2021), comprising subsets of iNaturalist (Van Horn et al., 2018), SUN(Xiao et al., 2010), PLACES(Zhou et al., 2017), and TEXTURE (Cimpoi et al., 2014). Crucially, the categories in each OOD dataset are non-overlapping with the ID dataset.

Additionally, we employ subsets of ImageNet-1k for fine-grained analysis. For example, we create ImageNet-10 to mirror the class distribution of CIFAR-10 but with high-resolution images. To further intensify the OOD evaluation, we curate ImageNet-20, consisting of 20 classes semantically akin to ImageNet-10, following (Ming et al., 2022). Regarding MS-COCO Lin et al. (2014), we follow the experiment setting presented by Miyai et al. (2023).

**Model:** We adopt the following configurations in our experimental studies:

**1) Pre-trained Model Selection:** We adopt CLIP (Radford et al., 2021) as our pre-trained model, aligning with established baselines (Ming et al., 2022; Wang et al., 2023). CLIP is widely recognized for its effectiveness in vision-language tasks and is accessible for various research purposes. It is important to note that while we utilize CLIP in our methodology, our approach is adaptable to other contrastive vision-language pre-training models emphasizing multi-modal feature alignment.

**2) Model Architecture:** Our primary model variant is CLIP-B/16, which utilizes a ViT-B/16 Transformer for the image encoder and a masked self-attention Transformer (Vaswani et al., 2017) for the text encoder. The input patch size in ViT models is denoted by "/x" in the model names, where "x" represents the patch resolution. We also introduce CLIP-L/14, based on ViT-L/14, as a representative of larger models.

**3) Threshold Setting:** For extracting general ID data representation, we set the threshold ($\epsilon_{th}$) in diminishing variance to $1e^{-4}$ for ImageNet-1k and MS-COCO as the ID datasets. For other ID datasets, the threshold is set to $1e^{-3}$. In section B.2 of Appendix, we report experimental results assessing the impact of the $k$ value on the performance of GRIC.

**4) PCA computation:** We select 25 random samples per class and run our PCA computation to find class-specific indices. We repeat the experiment for 10 runs, and report the experimental results in tables 1 and 2.

**5) Super-class Name Extraction:** To extract super-class names for ID datasets, we utilize a GPT3.5 (OpenAI, 2020) a large-language model capable of generating super-classes based on the provided ID labels (class names).

For more details on datasets and experimental configurations, please refer to the appendix, Section C.

| ID Dataset | Method | OOD Dataset | | | | | | | | Average | |
|---|---|---|---|---|---|---|---|---|---|---|---|
| | | iNaturalist | | SUN | | Places | | Texture | | | |
| | | FPR95↓ | AUROC↑ | FPR95↓ | AUROC↑ | FPR95↓ | AUROC↑ | FPR95↓ | AUROC↑ | FPR95↓ | AUROC↑ |
| CUB-200 (Welinder et al., 2010) | MCM | 9.83 | 98.24 | 4.93 | 99.10 | 6.65 | 98.57 | 6.97 | 98.75 | 7.09 | 98.66 |
| | GRIC | **4.10** | **99.14** | **3.08** | **99.46** | **2.98** | **99.06** | **3.14** | **99.17** | **3.33** | **99.21** |
| Stanford-Cars (Welinder et al., 2010) | MCM | 0.05 | 99.77 | 0.02 | 99.95 | 0.24 | 99.89 | 0.02 | 99.96 | 0.08 | 99.89 |
| | GRIC | **0.02** | **99.87** | **0.01** | **99.96** | **0.10** | **99.95** | **0.02** | **99.98** | **0.04** | **99.94** |
| FOOD-101 (Bossard et al., 2014) | MCM | 0.64 | 99.78 | 0.90 | 99.75 | 1.86 | 99.58 | 4.04 | 98.62 | 1.86 | 99.43 |
| | GRIC | **0.35** | **99.89** | **0.58** | **99.89** | **1.18** | **99.81** | **3.08** | **98.79** | **1.30** | **99.60** |
| Oxford-Pet (Parkhi et al., 2012) | MCM | 2.85 | 99.38 | 1.06 | 99.73 | 2.11 | 99.56 | **0.80** | **99.81** | 1.70 | 99.62 |
| | GRIC | **1.93** | **99.70** | **0.74** | **99.87** | **1.41** | **99.78** | 0.76 | 99.87 | **1.21** | **99.81** |
| ImageNet-10 (Deng et al., 2009) | MCM | 0.12 | 99.80 | 0.29 | 99.79 | 0.88 | 99.62 | **0.04** | **99.90** | 0.33 | 99.78 |
| | GRIC | **0.05** | **99.89** | **0.13** | **99.88** | **0.59** | **99.82** | 0.03 | 99.91 | **0.20** | **99.88** |
| ImageNet-20 (Deng et al., 2009) | MCM | 1.02 | 99.66 | 2.55 | 99.50 | 4.40 | 99.11 | 2.43 | 99.03 | 2.60 | 99.32 |
| | GRIC | **0.78** | **99.86** | **1.75** | **99.71** | **3.12** | **99.32** | **1.89** | **99.28** | **1.89** | **99.54** |
| ImageNet-100 (Deng et al., 2009) | MCM | 18.13 | 96.77 | 36.45 | 94.54 | 34.52 | 94.36 | 41.22 | 92.25 | 32.58 | 94.48 |
| | GRIC | **5.63** | **98.32** | **16.27** | **97.06** | **21.53** | **98.16** | **29.41** | **93.91** | **18.21** | **96.86** |
| MS-COCO (Lin et al., 2014) | MCM | 64.42 | 90.75 | 84.18 | 81.81 | NA | NA | 65.10 | 86.62 | 71.23 | 86.39 |
| | GL-MCM | 34.72 | 94.45 | 61.96 | 88.49 | NA | NA | 72.08 | 84.53 | 56.25 | 89.16 |
| | GRIC | **24.20** | **96.89** | **43.27** | **95.33** | NA | NA | **44.52** | **95.18** | **37.33** | **95.80** |

Table 1: Training-free multi-modal OOD detection based on CLIP-B/16 with various ID/OOD datasets. Notably, **GRIC exhibits superior performance**, surpassing other baseline methods in FPR95 and AUROC measures on average. The symbol ↑ denotes a preference for larger values, while ↓ indicates a preference for smaller values.

**Metrics:** In our comprehensive evaluation, we employ three well-known metrics in OOD detection to assess the performance of our approach. These metrics are:

**1) False Positive Rate at** $95\%$ **True Positive Rate (FPR95):** This metric measures the rate of false positives among OOD samples when the true positive rate of ID samples reaches $95\%$. It provides a nuanced assessment of the model's ability to discern between ID and OOD instances.

**2) Area Under the Receiver Operating Characteristic Curve (AUROC):** is a widely used metric that quantifies the model's ability to discriminate between in-distribution and out-of-distribution samples across various decision thresholds. A higher AUROC value indicates improved overall performance in distinguishing between the two classes.

**3) ID classification accuracy (ID ACC).** We provide the details in appendix, Section B.3.

## 4.2 EXPERIMENTAL RESULTS

1) **GRIC exhibits remarkable performance across diverse ID datasets:** Our evaluation highlights the efficacy of the GRIC method in training-free multi-modal OOD detection, utilizing a single pre-trained model across diverse tasks. We assessed GRIC across six ID datasets to demonstrate its adaptability and robustness in various scenarios, as summarized in Table 1. These results establish GRIC as a notably superior method for handling multi-task environments, specifically in OOD detection challenges.

In terms of FPR95, averaged over OOD datasets, GRIC consistently outperforms the MCM method. We observe substantial improvements with GRIC, achieving a decrease of $3.76\%$ on CUB-200, $0.04\%$ on Stanford-Cars, $0.56\%$ on FOOD-101, $0.49\%$ on Oxford-Pets, $0.13\%$ on ImageNet-10, $0.71\%$ on ImageNet-20, $14.37\%$ on ImageNet-100, and $18.92\%$ on MS-COCO. These results underscore GRIC's capability to significantly mitigate false detections across a spectrum of complex datasets.

Additionally, GRIC surpasses MCM in terms of AUROC, further averaged over OOD datasets. Here, GRIC shows enhancements of $0.55\%$ on CUB-200, $0.05\%$ on Stanford-Cars, $0.17\%$ on FOOD-101, $0.19\%$ on Oxford-Pets, $0.10\%$ on ImageNet-10, $0.22\%$ on ImageNet-20, and $2.38\%$ on ImageNet-100, and $33.90\%$ on MS-COCO. These improvements in AUROC confirm the superior detection accuracy of GRIC, highlighting its effectiveness in distinguishing between ID and OOD samples more reliably than MCM.

Our results confirm GRIC's robustness and versatility in training-free multi-modal OOD detection, highlighting its effectiveness in improving model reliability for real-world applications.

2) **GRIC demonstrates outstanding performance on large-scale ID data:** To comprehensively assess the scalability of GRIC, we conducted a thorough comparison with several recent and competitive OOD detection methods, including those proposed by (Ming et al., 2022; Wang et al., 2023; Fort et al., 2021; Huang & Li, 2021). Our evaluation focused on the widely used ImageNet-1k dataset (ID), with detailed results presented in Table 2. Our approach, GRIC, surpasses prompt-

| Method | OOD Dataset | | | | | | | | Average | |
|---|---|---|---|---|---|---|---|---|---|---|
| | iNaturalist (Van Horn et al., 2018) | | SUN (Xiao et al., 2010) | | Places (Zhou et al., 2017) | | Texture (Cimpoi et al., 2014) | | | |
| | FPR95↓ | AUROC↑ | FPR95↓ | AUROC↑ | FPR95↓ | AUROC↑ | FPR95↓ | AUROC↑ | FPR95↓ | AUROC↑ |
| Requires training (or w. fine-tuning) | | | | | | | | | | |
| ODIN (Liang et al., 2017) | 30.22 | 94.65 | 54.04 | 87.17 | 55.06 | 85.54 | 51.67 | 87.85 | 47.75 | 88.80 |
| MOS (Huang & Li, 2021) (BiT) | 9.28 | 98.15 | 40.63 | 92.01 | 49.54 | 89.06 | 60.43 | 81.23 | 39.97 | 90.11 |
| ViM (Wang et al., 2022a) | 32.19 | 93.16 | 54.01 | 87.19 | 60.67 | 83.75 | 53.94 | 87.18 | 50.20 | 87.82 |
| KNN (Sun et al., 2022) | 29.17 | 94.52 | 35.62 | 92.67 | 39.61 | 91.02 | 64.35 | 85.67 | 42.19 | 90.97 |
| NPOS$_{MCM}$ (Tao et al., 2023)$^+$ | 16.58 | 96.19 | 43.77 | 90.44 | 45.27 | 89.44 | 46.12 | 88.80 | 37.93 | 91.22 |
| NPOS$_{MCM}$ (Tao et al., 2023)$^*$ | 19.59 | 95.68 | 48.26 | 89.70 | 49.82 | 88.77 | 51.12 | 87.58 | 42.20 | 90.43 |
| Fort et al. (Fort et al., 2021) (ViT-B) | 15.07 | 96.64 | 54.12 | 86.37 | 57.99 | 85.24 | 53.32 | 84.77 | 45.12 | 88.25 |
| Fort et al. (Fort et al., 2021) (ViT-L) | 15.74 | 96.51 | 52.34 | 87.32 | 55.14 | 86.48 | 51.38 | 85.54 | 43.65 | 88.96 |
| Energy (Liu et al., 2020) (CLIP-B) | 21.59 | 95.99 | 34.28 | 93.15 | 36.64 | 91.82 | 51.18 | 88.09 | 35.92 | 92.26 |
| Energy (Liu et al., 2020) (CLIP-L) | 10.62 | 97.52 | 30.46 | 93.83 | 32.25 | 93.01 | 44.35 | 89.64 | 29.42 | 93.50 |
| MSP (Hendrycks & Gimpel, 2017) (CLIP-B) | 40.89 | 88.63 | 65.81 | 81.24 | 67.90 | 80.14 | 64.96 | 78.16 | 59.89 | 82.04 |
| MSP (Hendrycks & Gimpel, 2017) (CLIP-L) | 34.54 | 92.62 | 61.18 | 83.68 | 59.86 | 84.10 | 59.27 | 82.31 | 53.71 | 85.68 |
| CLIPN-C (Wang et al., 2023) (CLIP-B) | 28.58 | 90.88 | 31.64 | 89.38 | 56.59 | 78.28 | 37.55 | 86.85 | 38.59 | 86.35 |
| CLIPN-A (Wang et al., 2023)(CLIP-B) | 23.94 | 95.27 | 26.17 | 93.93 | 40.83 | 90.93 | 33.45 | 92.28 | 31.10 | 93.10 |
| Prompt learning | | | | | | | | | | |
| CoOp$_{MCM}$(Zhou et al., 2022b) | 43.38 | 91.26 | 38.53 | 91.95 | 46.68 | 89.09 | 50.64 | 87.83 | 44.81 | 90.03 |
| LoCoOp$_{MCM}$(Miyai et al., 2024) | 38.49 | 92.49 | 33.27 | 93.67 | 39.23 | 91.07 | 49.25 | 89.13 | 40.17 | 91.53 |
| NegPrompts (Li et al., 2024) | 6.32 | 98.73 | 22.89 | 95.55 | 27.60 | 93.34 | 35.21 | 91.60 | 23.01 | 94.81 |
| IDPrompt (Bai et al., 2024) (CLIP-B) | 8.98 | 98.19 | 42.03 | 91.64 | 44.00 | 90.57 | 25.27 | 94.32 | 26.08 | 94.36 |
| Training free | | | | | | | | | | |
| Entropy (CLIP-B) | 84.44 | 63.50 | 93.79 | 62.54 | 94.10 | 64.15 | 97.16 | 58.98 | 92.37 | 62.29 |
| Var (CLIP-B) | 87.42 | 63.87 | 68.71 | 81.02 | 76.28 | 75.38 | 80.04 | 71.90 | 78.11 | 73.04 |
| Scaled (CLIP-B) | 89.06 | 72.26 | 89.06 | 70.81 | 89.08 | 69.66 | 89.56 | 68.17 | 89.19 | 70.22 |
| MCM (CLIP-B) (Ming et al., 2022) | 30.91 | 94.61 | 37.59 | 92.57 | 44.69 | 89.77 | 57.77 | 86.11 | 42.74 | 90.77 |
| MCM (CLIP-L) (Ming et al., 2022) | 28.38 | 94.95 | 29.00 | 94.14 | 35.42 | 92.00 | 59.88 | 84.88 | 38.17 | 91.49 |
| GL-MCM (Miyai et al., 2023) | 15.18 | 96.71 | 30.42 | 93.09 | 38.85 | 89.90 | 57.93 | 83.63 | 35.47 | 90.83 |
| NegLabl (Jiang et al., 2024) | 1.91 | 99.49 | 20.53 | 95.49 | 35.59 | 91.64 | 43.56 | 90.22 | 25.40 | 94.21 |
| EOE (Cao et al., 2024) | 12.29 | 97.52 | 20.40 | 95.73 | 30.16 | 92.95 | 57.53 | 85.64 | 30.09 | 92.96 |
| GRIC (ours) (CLIP-B) | 10.32$^{\pm}$0.23 | 98.81$^{\pm}$0.10 | 20.11$^{\pm}$0.28 | 97.59$^{\pm}$0.14 | 24.37$^{\pm}$0.31 | 96.82$^{\pm}$0.29 | 26.51$^{\pm}$0.11 | 93.97$^{\pm}$0.25 | 20.32$^{\pm}$0.23 | 96.80$^{\pm}$0.20 |
| GRIC (ours) (CLIP-L) | 8.74$^{\pm}$0.22 | 99.12$^{\pm}$0.12 | 17.83$^{\pm}$0.21 | 98.06$^{\pm}$0.18 | 22.17$^{\pm}$0.18 | 97.51$^{\pm}$0.20 | 21.67$^{\pm}$0.14 | 95.14$^{\pm}$0.12 | 17.60$^{\pm}$0.19 | 97.45$^{\pm}$0.16 |

Table 2: Results for out-of-distribution (OOD) detection utilizing ImageNet-1k as the in-distribution (ID) dataset demonstrate the **remarkable performance of GRIC across various datasets**.

based methods that incorporate supervised training aspects, as demonstrated in prior works such as (Zhou et al., 2022a; Miyai et al., 2024). The table includes alternative scoring functions such as entropy, variance, and scaled, as per Ming et al. (Ming et al., 2022). For more details regarding our experiments settings, readers are encouraged to refer to the appendix, specifically Section C.

Utilizing CLIP-B as the pre-trained model and incorporating iNaturalist (Van Horn et al., 2018), SUN (Xiao et al., 2010), Places (Zhou et al., 2017), and Textures (Cimpoi et al., 2014) as OOD datasets, GRIC consistently outperforms competitors such as MCM and GL-MCM (Miyai et al., 2023).

The results in Table 2 demonstrate that GRIC (CLIP-B) significantly outperforms MCM and GL-MCM across all evaluated OOD datasets. In terms of FPR95, GRIC achieves consistently lower false positive rates, such as $10.32\%$ on iNaturalist and $26.51\%$ on Texture, compared to MCM ($30.91\%$ and $57.77\%$) and GL-MCM ($15.18\%$ and $57.93\%$). These improvements highlight GRIC's robustness in minimizing false positives, particularly on challenging datasets like Texture.

Additionally, GRIC achieves higher AUROC scores across all datasets, demonstrating superior detection accuracy. For instance, GRIC reports $98.81\%$ and $93.97\%$ on iNaturalist and Texture, respectively, outperforming MCM ($94.61\%$, $86.11\%$) and GL-MCM ($96.71\%$, $83.63\%$). This highlights GRIC's consistent performance across diverse OOD settings, establishing it as a state-of-the-art method for OOD detection with CLIP-B.

3) **GRIC demonstrates superior performance with larger models:** the model size comparison between CLIP-B and CLIP-L versions of GRIC highlights a clear performance improvement with increased model capacity. GRIC (CLIP-L) achieves even lower FPR95 and higher AUROC across all datasets, reflecting the positive impact of model scaling on both reducing false positives and improving overall detection reliability.

It is worth noting that CLIPN-C, CLIPN-A, and MOS (Huang & Li, 2021) display competitive performance on ImageNet-1k, albeit requiring model training or fine-tuning. In contrast, GRIC (CLIP-B) surpasses CLIPN-C and MOS by $18.27\%$ and $19.65\%$ in FPR95, highlighting its training-free advantage. These results highlight GRIC's robustness and effectiveness, especially where training-free approaches outperform methods needing additional training or fine-tuning.

Further detailed experiments are provided in the Appendix. These encompass the evaluation of GRIC performance utilizing a ResNet-based CLIP model, the analysis of the significance of the parameter K (representing the number of masked class-specific features) in GRIC performance, the assessment of ID classification accuracy, the impact of GRIC masking on single-modal OOD detection methods, and and the experiment considering masking one class at a time.

### 4.3 ABLATION STUDIES

As detailed in Section 3, our approach incorporates two key elements: the general representation of ID data and the use of informative prompts. This experiment aims to assess the individual impact of each factor on the performance of GRIC, with results summarized in Table 3.

| Method | OOD Dataset | | | | | | | | Average | |
|---|---|---|---|---|---|---|---|---|---|---|
| | iNaturalist (Van Horn et al., 2018) | | SUN (Xiao et al., 2010) | | Places (Zhou et al., 2017) | | Texture (Cimpoi et al., 2014) | | | |
| | FPR95↓ | AUROC↑ | FPR95↓ | AUROC↑ | FPR95↓ | AUROC↑ | FPR95↓ | AUROC↑ | FPR95↓ | AUROC↑ |
| GRIC | $10.32^{\pm0.23}$ | $98.81^{\pm0.10}$ | $20.11^{\pm0.28}$ | $97.59^{\pm0.14}$ | $24.37^{\pm0.31}$ | $96.82^{\pm0.29}$ | $26.51^{\pm0.11}$ | $93.97^{\pm0.25}$ | $20.32^{\pm0.23}$ | $96.80^{\pm0.20}$ |
| GRIC-IG, no IP | $16.27^{\pm0.31}$ | $96.38^{\pm0.28}$ | $24.49^{\pm0.18}$ | $95.11^{\pm0.43}$ | $29.38^{\pm0.20}$ | $93.75^{\pm0.09}$ | $34.49^{\pm0.42}$ | $90.28^{\pm0.21}$ | $26.16^{\pm0.28}$ | $93.88^{\pm0.25}$ |
| GRIC-IP, no ID General Rpresentation | 23.20 | 96.14 | 30.27 | 93.32 | 37.07 | 91.12 | 41.41 | 88.19 | 32.99 | 92.19 |
| GRIC [ Masked **Non-PCA** indices, w/o IP] | $39.11^{\pm0.38}$ | $85.08^{\pm0.21}$ | $43.19^{\pm0.36}$ | $85.14^{\pm0.19}$ | $58.02^{\pm0.46}$ | $77.23^{\pm0.29}$ | $62.88^{\pm0.27}$ | $80.31^{\pm0.14}$ | $50.80^{\pm0.37}$ | $81.95^{\pm0.21}$ |
| GRIC [ Masked **Non-PCA** indices + IP] | $36.67^{\pm0.29}$ | $88.81^{\pm0.22}$ | $40.61^{\pm0.34}$ | $87.30^{\pm0.10}$ | $55.17^{\pm0.23}$ | $78.47^{\pm0.17}$ | $61.55^{\pm0.34}$ | $83.71^{\pm0.24}$ | $48.50^{\pm0.30}$ | $84.57^{\pm0.18}$ |
| GRIC12 [12 samples/class] (CLIP-B) | $10.46^{\pm0.13}$ | $98.76^{\pm0.11}$ | $20.43^{\pm0.17}$ | $97.38^{\pm0.10}$ | $24.71^{\pm0.09}$ | $96.73^{\pm0.08}$ | $27.11^{\pm0.32}$ | $93.65^{\pm0.15}$ | $20.68^{\pm0.13}$ | $96.63^{\pm0.11}$ |
| GRIC16 [16 samples/class] (CLIP-B) | $10.41^{\pm0.12}$ | $98.79^{\pm0.08}$ | $20.40^{\pm0.13}$ | $97.39^{\pm0.15}$ | $24.67^{\pm0.12}$ | $96.75^{\pm0.17}$ | $27.01^{\pm0.16}$ | $93.67^{\pm0.11}$ | $20.62^{\pm0.13}$ | $96.65^{\pm0.13}$ |
| GL-MCM + informative prompts | 14.11 | 97.08 | 30.27 | 93.73 | 38.17 | 90.14 | 51.38 | 86.48 | 33.48 | 91.86 |

Table 3: Ablation study on GRIC. Both **ID data general representation** and **informative prompts** play important roles on promising performance of GRIC. IP stands for informative prompts. We leverage CLIP-B and ImageNet-1k as our pre-trained model and ID dataset.

1) **ID Data general representation empowers GRIC:** In this study, we conducted experiments under the same settings as presented in Table 2, utilizing simplified prompts (e.g., "a photo of a $\langle y_i \rangle$") without providing any super-class information. This variant of GRIC is referred to as GRIC-IG. Table 3 illustrates that leveraging a general representation of ID data significantly enhances performance compared to the MCM. Regarding FPR95, GRIC-IG outperforms iNaturalist, SUN, Places, and Textures by $14.64\%$, $13.10\%$, $15.31\%$, and $23.28\%$, respectively. In AUROC, GRIC-IG exhibits an average improvement of $3.11\%$ over MCM across the four OOD datasets.

2) **Informative Prompts enhances GRIC:** In this experiment, we maintained the experimental settings from Table 2, employing the full representation of the data without masking any features. This variant is denoted as GRIC-IP. Table 3 underscores the crucial role of informative prompts in achieving outstanding performance for GRIC. In terms of FPR95, GRIC-IP outperforms iNaturalist, SUN, Places, and Textures by $7.71\%$, $7.32\%$, $7.62\%$, and $16.36\%$, respectively. Considering AUROC, GRIC-IP exhibits an average improvement of $1.42\%$ over MCM. This analysis underscores the synergistic impact of both ID data general representation and informative prompts, demonstrating their contributions to the effectiveness of GRIC in training-free multi-modal OOD detection.

3) **Masking Non-PCA indices:** We evaluate the impact of masking the most important (class-specific) features versus less important (general, Non-PCA) features on our method's performance. Specifically, we retain the top $k$ important features while masking the rest. Experiments are conducted with and without informative prompts (IP). Table 3 shows that masking Non-PCA features degrades performance in both cases, highlighting their critical role in maintaining overall effectiveness, irrespective of IP usage.

4) **PCA computation with limited data**: Our approach is notably label-free and does not involve training on ID or OOD data. However, it should be noted that the method requires access to a small amount of ID data solely for a one-time PCA computation, positioning it as a few-shot method. In the current experiment, we demonstrate that our method achieves strong performance even when accessing a limited number of ID samples. Specifically, PCA was computed using only 12-shot per class (resulting in a matrix of dimensions [121000, 512]) and 16-shot per class (resulting in a matrix of dimensions [161000, 512]). Importantly, the PCA computation is conducted post-forward pass, utilizing the extracted features, rather than during the training process. As a result, our approach incurs significantly lower computational costs compared to training-based methods, further reinforcing its efficiency and practicality.

## 5 CONCLUSION

In conclusion, this research study addresses the critical challenge of out-of-distribution sample detection in machine learning applications by introducing GRIC leveraging general representation of the ID data and informative content. By emphasizing general feature representations within the in-distribution data and employing informative prompts enriched with ID super-class information, GRIC showcases its efficacy in nuanced OOD detection. Our contributions include the introduction of GRIC as a novel and effective technique, which, through comprehensive evaluations, outperforms various OOD detection baselines, achieving an average reduction of $\sim 19\%$ in FPR95 compared to robust baseline methods. These findings offer valuable insights into the intricacies and efficacy of GRIC, positioning it as a promising solution for advancing training-free OOD detection in multi-modal machine learning applications.

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

# A  Appendix

This supplementary material provides in-depth information on the following topics:

- Additional Experiments.
- Experiment Details.
- Related Works.
- Training-Based Multi-modal Out-of-Distribution (OOD) Detection Methods
- Principal Component Analysis (PCA).
- Mean Variance Calculation.

Each section offers detailed insights into the respective topic for a comprehensive understanding.

# B  Additional Experiments

## B.1  GRIC Leveraging ResNet-based CLIP Models

Our primary findings are based on the CLIP model featuring a Vision Transformer (ViT) image encoder. Additionally, we explore the efficacy of GRIC for models based on ResNet architecture in the context of CLIP. Specifically, we employ the ResNet model with a depth of 50 and a width multiplier of 4 (RN50x4) with 178.3 million parameters, a parameter count comparable to CLIP-B/16 (149.6 million). The results are presented in Table 4.

The outcomes demonstrate that GRIC continues to yield promising results when applied to ResNet-based CLIP models. The performance remains competitive between RN50x4 and CLIP-B/16, with AUROC values of 90.68 and 92.89, respectively.

| Method | OOD Dataset | | | | | | | | Average | |
| | iNaturalist (Van Horn et al., 2018) | | SUN (Xiao et al., 2010) | | Places (Zhou et al., 2017) | | Texture (Cimpoi et al., 2014) | | | |
| | FPR95↓ | AUROC↑ | FPR95↓ | AUROC↑ | FPR95↓ | AUROC↑ | FPR95↓ | AUROC↑ | FPR95↓ | AUROC↑ |
| --- | --- | --- | --- | --- | --- | --- | --- | --- | --- | --- |
| GRIC (ours) (RN50x4) | 38.80 | 92.03 | 32.93 | 93.23 | 38.11 | 90.00 | 49.96 | 87.49 | 39.95 | 90.68 |
| GRIC (ours) (CLIP-B/16) | $10.32^{\pm 0.23}$ | $98.81^{\pm 0.10}$ | $20.11^{\pm 0.28}$ | $97.59^{\pm 0.14}$ | $24.37^{\pm 0.31}$ | $96.82^{\pm 0.29}$ | $26.51^{\pm 0.11}$ | $93.97^{\pm 0.25}$ | $20.32^{\pm 0.23}$ | $96.80^{\pm 0.20}$ |
| MCM (RN50x4) | 44.51 | 91.51 | 35.11 | 92.84 | 43.74 | 89.60 | 57.73 | 85.93 | 45.27 | 89.97 |
| MCM (CLIP-B/16) | 30.91 | 94.61 | 37.59 | 92.57 | 44.69 | 89.77 | 57.77 | 86.11 | 42.74 | 90.77 |

Table 4: GRIC presents outstanding performance leveraging ResNet-based CLIP model on ImageNet-1k (ID).

## B.2  Evaluating the Significance of $k$ in GRIC Performance

As elucidated in Section 3, we assign a value of zero to the $k$ most important features to derive a general representation of in-distribution (ID) data. The determination of the $k$ value is accomplished through diminishing mean-variance, with a set threshold of $1e^{-4}$ in our ImageNet-1k experiment, resulting in an optimal $k$ value of $34$ to meet this threshold.

In this investigation, we systematically assess the impact of the $k$ value on the performance of GRIC by employing different $k$ values, specifically 30, 40, and 45. The experimental outcomes are detailed in Table 5. Notably, deviations from $k = 34$ demonstrate discernible effects on the method's performance. Generally, values close to $k = 34$ demonstrate discernible effects on the method's performance. Generally, values close to $k = 34$ exhibit comparable performance. However, as the deviation from $k = 34$ increases, there is a noticeable degradation in performance.

In conclusion, our analysis underscores the substantial influence of the $k$ parameter on the performance of the GRIC method. The selection of an appropriate $k$ value emerges as a critical factor in achieving optimal results across diverse out-of-distribution datasets.

## B.3  ID Classification Accuracy

To augment the precision of ID classification, we integrate the comprehensive feature representation, acknowledging its indispensable role in the identification process. Simultaneously, we incorporate informative prompts that leverage hierarchy information, aligning with the methodology employed in our out-of-distribution (OOD) detection experiments, denoting it as GRIC-IP. The outcomes of our experiments are detailed in Table 3, underscoring the exceptional performance achieved with

| Method | OOD Dataset | | | | | | | | Average | |
|---|---|---|---|---|---|---|---|---|---|---|
| | iNaturalist (Van Horn et al., 2018) | | SUN (Xiao et al., 2010) | | Places (Zhou et al., 2017) | | Texture (Cimpoi et al., 2014) | | | |
| | FPR95↓ | AUROC↑ | FPR95↓ | AUROC↑ | FPR95↓ | AUROC↑ | FPR95↓ | AUROC↑ | FPR95↓ | AUROC↑ |
| GRIC, $k = 34$ | 10.32 | 98.81 | 20.11 | 97.59 | 24.37 | 96.82 | 26.51 | 93.97 | 20.32 | 96.80 |
| GRIC, $k = 30$ | 10.67 | 98.66 | 20.78 | 97.09 | 24.70 | 96.56 | 26.84 | 93.80 | 20.75 | 96.53 |
| GRIC, $k = 40$ | 13.19 | 97.01 | 23.09 | 95.98 | 28.37 | 92.47 | 29.18 | 92.11 | 23.46 | 94.39 |
| GRIC, $k = 45$ | 13.68 | 96.81 | 23.43 | 95.82 | 28.19 | 92.35 | 29.53 | 91.93 | 23.71 | 94.23 |

Table 5: Impact of $k$ in performance of GRIC.

GRIC-IP. The results indicate that the incorporation of informative prompts contributes to an enhancement in the ID classification performance. Notably, it is crucial to emphasize that MCM represents the base of our methodology, omitting both the consideration of general ID representation and the utilization of informative prompts.

| Method | ID ACC |
|---|---|
| **Training free** | |
| MCM (CLIP-B/16) | 67.01 |
| MCM (CLIP-L/14) | 73.28 |
| GRIC-IP (CLIP-B/16) | 80.29 |
| GRIC-IP (CLIP-L/14) | 85.64 |
| **w. fine-tuning** | |
| MSP (CLIP-B/16) | 79.39 |
| MSP (CLIP-L/14) | 84.12 |
| Energy (Liu et al., 2020) (CLIP-B/16) | 79.39 |
| Energy (Liu et al., 2020) (CLIP-L/14) | 84.12 |
| Fort et al. (Fort et al., 2021) (ViT-B/16) | 81.25 |
| Fort et al. (Fort et al., 2021) (ViT-L/14) | 84.05 |
| MOS (Huang & Li, 2021) (BiT) | 75.16 |

Table 6: The accuracy of ID classification on ImageNet-1k (%) demonstrates promising performance with our method, GRIC-IP, which utilizes informative prompts.

Furthermore, Table 6 presents the multi-class classification accuracy on ImageNet-1k for the methods listed in Table 2.

### B.4   GRIC MASKING (GM) LEADS TO A NOTABLE ENHANCEMENT IN THE PERFORMANCE OF SINGLE-MODAL METHODS:

We conducted supplementary experiments to assess the influence of incorporating the general representation of in-distribution (ID) data on single-modal out-of-distribution (OOD) detection methodologies such as Mahalanobis (Lee et al., 2018), Energy score (Liu et al., 2020), React (Sun et al., 2021a), and GradNorm (Huang et al., 2021).

We utilize the ImageNet-1k dataset as the ID dataset in our experimental setup. Firstly, we compute mask indices and general feature representations of ID data from ImageNet-1k. Subsequently, we apply these mask indices to each test sample before subjecting them to traditional single-modal OOD detection methods. This methodology enables us to assess how leveraging general ID data representations influences the performance of OOD detection algorithms.

Results and Discussion: Our experimental results, as presented in Table 7 demonstrate that leveraging general feature representations from the ImageNet-1k dataset leads to improvements in the average AUROC performance of Mahalanobis, GradNorm, Energy score, and React OOD detection methods by $3.91$, $2.74$, $3.76$, and $0.31$, respectively. These findings highlight the significance of incorporating general ID data representations in enhancing the effectiveness of traditional single-modal OOD detection algorithms.

### B.5   MASKING ONE CLASS AT A TIME

In section 4.2, we present the initial findings from our experiments, focusing on the performance evaluation of our method across various ID datasets. The summarized outcomes are presented in Table 1. For this experiment, we derived masking indices and a general representation using all classes. An intriguing aspect of our approach involves masking class-specific features for individual classes. To delve deeper into this aspect, we conducted supplementary experiments, masking one class at a time while leveraging the ImageNet10 ID dataset.

| Single-modal method | FPR95↓ | AUROC↑ | Single-modal method+GM | FPR95↓ | AUROC↑ |
|---|---|---|---|---|---|
| Mahalanobis (Lee et al., 2018) | 87.43 | 55.47 | Mahalanobis + GM | **75.26** | **61.92** |
| GradNorm (Huang et al., 2021) | 40.29 | 87.34 | GradNorm + GM | **31.16** | **91.62** |
| Energy (Liu et al., 2020) | 58.41 | 86.17 | Energy + GM | **47.08** | **89.37** |
| React (Sun et al., 2021a) | 31.43 | 92.95 | React + GM | **25.31** | **95.73** |

Table 7: **GRIC Masking (GM) improves** most Single-modal methods significantly.

| Single-modal method | FPR95↓ | AUROC↑ | Single-modal method+GM | FPR95↓ | AUROC↑ |
|---|---|---|---|---|---|
| GRIC [All] | 0.20 | 99.88 | MCM | 0.33 | 99.78 |
| GRIC [car] | 0.31 | 99.73 | GRIC [bird] | 0.34 | 99.75 |
| GRIC [cat] | 0.29 | 99.80 | GRIC [antelope] | 0.32 | 99.76 |
| GRIC [dog] | 0.30 | 99.82 | GRIC [frog] | 0.31 | 99.77 |
| GRIC [truck] | 0.43 | 99.73 | GRIC [horse] | 0.32 | 99.75 |
| GRIC [warplane] | 0.38 | 99.69 | GRIC [Ship] | 0.40 | 99.61 |

Table 8: Masking one class at a time,ImageNet10 as ID. *x* refers to the masked class in GRIC [x].

Following the experiment reported in Table 1, we evaluated our method using four OOD datasets: iNaturalist, SUN, Places, and Textures. We report the average performance metrics over these datasets, considering FPR95 and AUROC.

The detailed experimental outcomes are presented in Table 8. As shown in Table 8, masking different classes affects the performance variably. We observed that the best performance was achieved when leveraging masking generated by considering all classes collectively. Furthermore, our results highlight the importance of specific classes, prompting further investigation into the optimal selection of classes for masking. However, this aspect falls beyond the scope of the current paper and warrants future research investigations.

## B.6 MASKING NON-PCA FEATURES

We conducted experiments to assess the effect of retaining PCA features while masking non-PCA features, both with and without the use of informative prompts. The results of these experiments are summarized in Table 9.

| Method | OOD Dataset | | | | | | | | | |
|---|---|---|---|---|---|---|---|---|---|---|
| | iNaturalist | | SUN | | Places | | Texture | | Average | |
| | FPR95↓ | AUROC↑ | FPR95↓ | AUROC↑ | FPR95↓ | AUROC↑ | FPR95↓ | AUROC↑ | FPR95↓ | AUROC↑ |
| GRIC [Masked Non-PCA indices, without Informative Prompts] | 33.11 | 87.08 | 35.21 | 89.26 | 47.31 | 81.46 | 55.09 | 85.17 | 42.68 | 85.74 |
| GRIC [Masked Non-PCA indices, with Informative Prompts] | 30.21 | 91.37 | 32.84 | 91.18 | 45.72 | 83.03 | 52.49 | 82.96 | 40.32 | 87.14 |
| GRIC | 10.32 | 98.81 | 20.11 | 97.59 | 24.37 | 96.82 | 26.51 | 93.97 | 20.32 | 96.80 |

Table 9: Ablation study on GRIC evaluating the impact of masking Non-PCA features.

The results in the table 9 highlight the effectiveness of different feature masking strategies for OOD detection using GRIC. Masking non-PCA features provides some improvement, particularly when combined with informative prompts, as seen in the reduced FPR95 and increased AUROC across datasets. However, the proposed GRIC approach, leveraging PCA-masked features with prompts, significantly outperforms other methods. It achieves an average FPR95 of 20.32 and AUROC of 96.80, demonstrating superior robustness and generalizability across diverse datasets such as iNaturalist, SUN, Places, and Texture. These findings validate the importance of PCA feature masking and informative prompts in enhancing OOD detection performance.

## B.7 GRIC ID ACCURACY WITHOUT INFORMATIVE PROMPTS

We evaluated GRIC's ID accuracy in the absence of informative prompts. The results are summarized in Table 10.

| Method | Model | ID ACC (%) |
|---|---|---|
| GRIC-IP | CLIP-B/16 | 80.29 |
| GRIC-IP | CLIP-L/14 | 85.64 |
| GRIC-No Prompts | CLIP-B/16 | 75.50 |
| GRIC-No Prompts | CLIP-L/14 | 78.59 |

Table 10: ID accuracy of GRIC with and without informative prompts.

With **Informative Prompts**, GRIC-IP achieves 80.29% accuracy using CLIP-B/16 and 85.64% using CLIP-L/14 by leveraging enriched textual prompts.

Without **Informative Prompts**, ID accuracy drops moderately by approximately 4-5%, highlighting the critical role of the masked features in distinguishing classes.

**Insight:** This moderate decline in accuracy validates that GRIC selectively removes class-specific features crucial for ID accuracy, demonstrating the importance of the masked feature spaces.

## B.8 HARD OOD DETECTION EXPERIMENT

Hard OOD detection tackles the challenge of identifying OOD samples that closely resemble ID samples, making the classification process more complex. In this study, we adopted the experimental setup described in NegPrompt (Li et al., 2024), using ImageNet-1K as the source for both ID and OOD samples.

To maintain consistency with previous research, we adhered to the predefined splits, ensuring a rigorous and diverse evaluation framework for challenging OOD scenarios. Details on the class distribution and the number of training/testing samples for each split can be found in Table 11 (Li et al., 2024).

The results in Table 12 highlight the strong performance of our few-shot approach in distinguishing ID and OOD samples, showcasing its reliability and robustness in addressing hard OOD detection tasks.

| Split | ID | OOD |
|---|---|---|
| Split-1 | All dog classes116: 1856 / 5800 | Non-animal classes166: — / 8300 |
| Split-2 | Half of hunting dog classes30: 480 / 1500 | Other 4-legged animal classes55: — / 2750 |
| Split-3 | Mix of common classes151: 2416 / 7550 | Mix of common classes164: — / 8200 |
| Split-4 | First 100 classes100: 1600 / 5000 | Remaining 900 classes900: — / 45000 |

Table 11: ImageNet-1K splits for hard OOD detection. Given are the numbers of *classes : training / test samples* following (Li et al., 2024).

| Method | Split-1 | | Split-2 | | Split-3 | | Split-4 | | Avg | |
|---|---|---|---|---|---|---|---|---|---|---|
| | AUC↑ | FPR95↓ | AUC↑ | FPR95↓ | AUC↑ | FPR95↓ | AUC↑ | FPR95↓ | AUC↑ | FPR95↓ |
| Zero/Few-shot methods | | | | | | | | | | |
| MCM (Ming et al., 2022) | 97.93 | 9.17 | 88.10 | 56.40 | 90.34 | 33.05 | 98.72 | 4.73 | 93.77 | 25.83 |
| CLIPN (Wang et al., 2023) | 99.38 | 2.07 | 97.77 | 10.55 | 90.03 | 36.85 | 98.83 | 4.68 | 96.50 | 13.53 |
| GRIC | 99.42 | 2.00 | 98.10 | 8.50 | 94.85 | 21.90 | 96.90 | 10.25 | 97.07 | 10.66 |
| CLIP-based posthoc methods | | | | | | | | | | |
| MSP (Hendrycks & Gimpel, 2017) | 77.85 | 63.60 | 68.73 | 83.63 | 79.10 | 70.55 | 82.40 | 65.52 | 77.02 | 70.83 |
| MaxLogit (Hendrycks et al., 2019) | 99.87 | 0.49 | 98.06 | 8.69 | 90.96 | 34.34 | 99.35 | 2.66 | 97.06 | 11.55 |
| Energy | 99.88 | 0.46 | 98.18 | 8.40 | 90.65 | 35.02 | 99.36 | 2.83 | 97.02 | 11.68 |
| ReAct (Sun et al., 2021a) | 99.34 | 0.72 | 97.91 | 9.33 | 90.85 | 35.65 | 99.12 | 2.94 | 96.80 | 12.16 |
| ODIN (Liang et al., 2017) | 98.78 | 1.12 | 98.23 | 8.18 | 89.92 | 37.20 | 98.76 | 13.20 | 96.42 | 14.92 |
| Prompt learning methods | | | | | | | | | | |
| CoOp (Zhou et al., 2022b) | 98.53 | 6.78 | 88.25 | 50.76 | 90.64 | 33.89 | 98.54 | 5.11 | 93.99 | 24.14 |
| LoCoOp (Miyai et al., 2024) | 98.64 | 6.29 | 84.63 | 61.09 | 91.30 | 27.79 | 98.83 | 41.44 | 93.35 | 34.15 |
| NegPrompt (Li et al., 2024) | 99.85 | 0.62 | 98.54 | 7.60 | 93.89 | 22.89 | 99.57 | 1.60 | 97.96 | 8.18 |
| Open-vocabulary OOD detection | | | | | | | | | | |
| CoOp (10%) (Zhou et al., 2022b) | 97.97 | 12.217 | 80.11 | 74.62 | 87.92 | 46.00 | 96.59 | 16.60 | 90.65 | 37.36 |
| LoCoOp (10%) (Miyai et al., 2024) | 98.00 | 9.23 | 87.02 | 46.02 | 90.82 | 32.84 | 98.24 | 48.72 | 86.99 | 42.52 |
| NegPrompt (10%) (Li et al., 2024) | 99.66 | 1.36 | 96.30 | 19.89 | 91.75 | 26.92 | 99.46 | 5.24 | 96.46 | 13.36 |

Table 12: Hard OOD detection results.

## C EXPERIMENT DETAILS

The results in the table highlight the effectiveness of different feature masking strategies for OOD detection using GRIC. Masking non-PCA features provides some improvement, particularly when combined with informative prompts, as seen in the reduced FPR95 and increased AUROC across datasets. However, the proposed GRIC approach, leveraging PCA-masked features with prompts, significantly outperforms other methods. It achieves an average FPR95 of 33.90 and AUROC of 92.89, demonstrating superior robustness and generalizability across diverse datasets such as iNaturalist, SUN, Places, and Texture. These findings validate the importance of PCA feature masking and informative prompts in enhancing OOD detection performance.

### C.1 SOFTWARE AND HARDWARE

**Software** We run all experiments with Python 3.8.0 and PyTorch 1.12.1.

**Hardware** All experiments are run on NVIDIA RTX 3090.

## C.2 HYPERPARAMETERS

As we explained in section 3.3, the formal definition of the matching score $S(x; \mathcal{Y}in, \mathcal{T}, \mathcal{I})$ is given by:

$$S(x) = \max_i \frac{e^{s_i(x)/\tau}}{\sum_{j=1}^{N} e^{s_j(x)/\tau}}, \tag{8}$$

where we set $\mathcal{T}$ to 1 in our formulation. The sole hyper parameter governing our model is the temperature scaling factor denoted as $\tau$. Our empirical investigations indicate that, our scoring function exhibits robustness to variations in the scaling factor. Specifically, across a broad range of values spanning from 0.5 to 100, the performance remains consistent.

## C.3 DATASETS

**ImageNet-10** We establish ImageNet-10, designed to emulate the class distribution of CIFAR-10, while utilizing high-resolution images. This dataset encompasses the following categories, each accompanied by its respective class ID: warplane (n04552348), sports car (n04285008), brambling bird (n01530575), Siamese cat (n02123597), antelope (n02422699), Swiss mountain dog (n02107574), bullfrog (n01641577), garbage truck (n03417042), horse (n02389026), and container ship (n03095699).

**ImageNet-20** For rigorous out-of-distribution (OOD) evaluation using realistic datasets, we adopt ImageNet-20, a dataset introduced by MCM. ImageNet-20 is meticulously curated, comprising 20 classes that are semantically akin to those in ImageNet-10, such as dog (in-distribution) versus wolf (OOD). The selection of categories is based on the semantic distance in the WordNet synsets (Fellbaum, 2010). The dataset encompasses the following categories: sailboat (n04147183), canoe (n02951358), balloon (n02782093), tank (n04389033), missile (n03773504), bullet train (n02917067), starfish (n02317335), spotted salamander (n01632458), common newt (n01630670), zebra (n01631663), frilled lizard (n02391049), green lizard (n01693334), African crocodile (n01697457), Arctic fox (n02120079), timber wolf (n02114367), brown bear (n02132136), moped (n03785016), steam locomotive (n04310018), space shuttle (n04266014), and snowmobile (n04252077). The generation of this dataset is facilitated using the script provided by the authors of MCM.

**ImageNet-100** We compile a dataset named ImageNet-100 by selecting 100 classes from ImageNet-1k. The MCM authors randomly chose these 100 classes without adhering to specific criteria. The dataset creation process is executed using the script provided by the MCM authors. The list of classes utilized in this dataset is accessible at `https://github.com/deeplearning-wisc/MCM`.

**Conventional Out-of-Distribution (OOD) Datasets** Huang et al.(Huang & Li, 2021) meticulously compile a diverse set of subsets from prominent datasets such as iNaturalist(Van Horn et al., 2018), SUN (Xiao et al., 2010), Places (Zhou et al., 2017), and Texture (Cimpoi et al., 2014), establishing expansive OOD datasets for ImageNet-1k. Importantly, the test sets for these datasets are designed such that their classes do not overlap with those in ImageNet-1k. A brief overview of each dataset is provided below.

**iNaturalist**: Comprising images captured in the natural world (Van Horn et al., 2018), iNaturalist boasts 13 super-categories and 5,089 sub-categories, spanning various domains such as plants, insects, birds, mammals, and more. For our purposes, we utilize a subset encompassing 110 plant classes that do not overlap with those present in ImageNet-1k.

**SUN**: An acronym for the Scene Understanding Dataset (Xiao et al., 2010), SUN encompasses 899 categories, encapsulating diverse indoor, urban, and natural environments, both with and without human presence. We selectively use a subset of 50 categories representing natural objects absent in ImageNet-1k.

**Places**: As a repository of large-scale scene photographs (Zhou et al., 2017), Places categorizes images into Indoor, Nature, and Urban scenes. From the larger collection, we extract a subset comprising 50 categories that are distinct from those found in ImageNet-1k.

**Texture**: Denoting the Describable Textures Dataset (Cimpoi et al., 2014), Texture consists of images featuring textures and abstracted patterns. Given the absence of category overlaps with

ImageNet-1k, we utilize the entire dataset, aligning with the approach taken by Huang et al. (Huang & Li, 2021).

### C.4 BASELINE MODELS AND MODEL CHECKPOINT SOURCES

In our evaluation of baseline models, we rely on reported experimental results from MCM (Ming et al., 2022) and CLIPN (Wang et al., 2023). For the Mahalanobis score (Lee et al., 2018), we utilize feature embeddings without $l_2$ normalization, considering that Gaussian distributions are inherently incompatible with hyperspherical features. Alternatively, one can opt to normalize the embeddings before applying the Mahalanobis score.

In the case of Fort et al. (2021), detailed in Table2, the entire Vision Transformer (ViT) model undergoes fine-tuning on the in-distribution (ID) dataset. We leverage publicly available checkpoints from Hugging Face, where the ViT model is pre-trained on ImageNet-21k and subsequently fine-tuned on ImageNet-1k. For instance, the checkpoint for ViT-B can be accessed at `https://huggingface.co/google/vit-base-patch16-224`.

Regarding CLIP models, our reported results are based on checkpoints provided by Hugging Face for CLIP-B (`https://huggingface.co/openai/clip-vit-base-patch16`) and CLIP-L (`https://huggingface.co/openai/clip-vit-large-patch14`). Similar outcomes can be achieved using checkpoints available in the OpenAI codebase (`https://github.com/openai/CLIP`). Notably, for CLIP (RN50x4), which is not accessible via Hugging Face, we employ the checkpoint provided directly by OpenAI.

## D  RELATED WORKS

**Vision-Language Models.**  The usage of large-scale pre-trained vision-language models for multimodal tasks has emerged as a promising paradigm, exhibiting impressive performance (Gu et al., 2020). Typically, two architectural paradigms are prevalent: single-stream models, exemplified by VisualBERT (Li et al., 2019a) and ViLT (Kim et al., 2021), which integrate text and visual features into a single transformer-based encoder; and dual-stream models like CLIP (Radford et al., 2021), ALIGN (Jia et al., 2021), and FILIP (Yao et al., 2021), employing separate encoders for text and image. These models optimize with contrastive objectives to align semantically similar features across different modalities. Among these, CLIP has gained widespread popularity due to its simplicity and robust performance. The success of CLIP-like models has prompted subsequent works (Li et al., 2022; Zhang et al., 2021), focusing on enhancing data efficiency and task adaptation. While our paper centers around CLIP as the primary pre-trained model, the proposed approach can generally apply to contrastive models aiming to align vision and language features.

Here's the combined and revised paragraph:

—

**OOD Detection in Computer Vision.**  For open-world multi-class classification, the objective of OOD detection is to establish a binary ID-OOD classifier alongside a multi-class model tailored for visual inputs. Various methodologies have emerged for deep neural networks (Yang et al., 2021b). These approaches include generative model-based techniques (Cai & Li, 2023; Ge et al., 2017; Kirichenko et al., 2020), as well as discriminative-model based methods. Within the latter category, OOD scores are derived from the model's softmax output (DeVries & Taylor, 2018; Hein et al., 2019; Yang et al., 2021a), energy-based scores (Liu et al., 2020; Sun et al., 2021b; Sun & Li, 2022), or gradient information (Behpour et al., 2023; Huang et al., 2021). Theoretical analyses have been presented by (Morteza & Li, 2022; Fang et al., 2022; Bitterwolf et al., 2022) in the domain of OOD detection.

Recent works (Roy et al., 2022; Wang et al., 2022b) have explored OOD detection specifically in long-tailed distributions. So far, these works have primarily concentrated on task-specific models using only visual information. Our method marks a pioneering leap in training-free multi-modal OOD detection, incorporating informative textual information alongside the shared general visual representation within in-distribution data across a spectrum of diverse tasks. CSP (Chen et al., 2024) and NegLabel Jiang et al. (2024) are advanced OOD detection methods leveraging vision-language models (VLMs) with distinct strategies:

CSP (Chen et al., 2024) employs a 'conjugated semantic pool' of superclasses to broaden semantic diversity, reducing reliance on specific labels and improving OOD detection for diverse cases. NegLabel Jia et al. (2021) introduces 'negative labels' selected from a large corpus to enhance semantic separability between ID and OOD samples, achieving exceptional robustness to domain shifts.

# E TRAINING-BASED MULTI-MODAL OUT-OF-DISTRIBUTION (OOD) DETECTION METHODS

In accordance with the discussions presented in Section D, numerous studies have explored the realm of multi-modal OOD detection, employing various training strategies.

**CLIPN: Saying "No" with CLIP** Wang et al. (2023) introduce CLIPN, an extension of CLIP (Contrastive Language-Image Pre-training) specifically designed for discerning between ID and OOD samples. CLIPN achieves this by incorporating positive semantic prompts and introducing negation-semantic prompts. The method employs a learnable "no" prompt and a dedicated "no" text encoder to capture negation semantics within images. Dual loss functions, the image-text binary-opposite loss, and the text semantic-opposite loss, are utilized to instruct CLIPN in associating images with "no" prompts, enabling it to identify unknown samples effectively.

**ZOC: Zero-Shot OOD Detection based on CLIP** Esmaeilpour et al. (Esmaeilpour et al., 2022) present the Zero-Shot OOD Detection (ZOC) method, extending the pre-trained language-vision model CLIP. ZOC incorporates a text-based image description generator trained on top of CLIP. During testing, this extended model generates candidate unknown class names for each test sample. A confidence score is then computed based on both known class names and candidate unknown class names, facilitating zero-shot OOD detection.

**CLIPood: Generalizing CLIP to OOD Test Data**

Shu et al. (Shu et al., 2023) proposed CLIPood, a fine-tuning method aimed at adapting CLIP models to out-of-distribution scenarios in downstream tasks. CLIPood addresses situations involving domain shifts and open classes in unseen test data. Introducing the margin metric softmax (MMS) as a novel training objective, CLIPood exploits semantic relations between classes from the text modality. Additionally, it incorporates a new optimization strategy, Beta moving average (BMA), for maintaining a temporal ensemble weighted by a Beta distribution. The focus of our paper centers on few-shot multi-modal OOD detection, and thus, studies involving training a text encoder, such as those discussed above, fall outside the scope of our investigation.

In summary, these training-based methods showcase diverse approaches to multi-modal OOD detection, each contributing unique insights and methodologies. However, our emphasis in this paper is specifically on few-shot multi-modal OOD detection, excluding investigations that involve training a text encoder.

# F PRINCIPAL COMPONENT ANALYSIS (PCA)

PCA aims to transform high-dimensional data into a lower-dimensional representation while retaining the maximum variance in the data. It achieves this by identifying the principal components, which are orthogonal vectors that capture the directions of maximum variance (Shlens, 2014; Jolliffe, 2002).

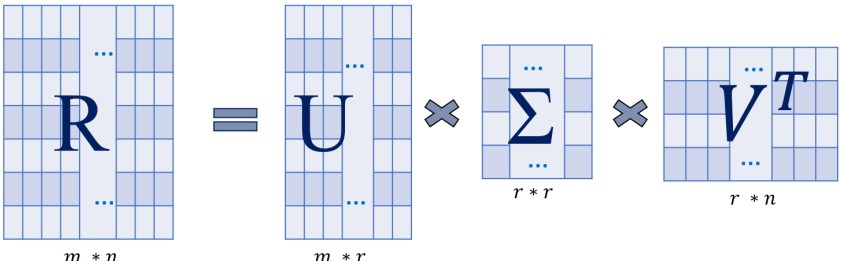

Figure 4: Singular Value Decomposition

### F.1 Procedure:

In this section, we explain the procedure that is followed in PCA analysis to extract the most informative features.

**Data Standardization:**

Standardize the features of the dataset to have zero mean and unit variance.

**Covariance Matrix:** Compute the covariance matrix of the standardized data. The covariance matrix represents the relationships between different features.

**SVD of Covariance Matrix:**

Perform SVD on the covariance matrix. The singular value decomposition of the covariance matrix results in the principal components. Selecting Principal Components:

Sort the singular values in descending order. The corresponding singular vectors are the principal components. Choose the top k principal components to form a reduced-dimensional space.

**Projection:**

Project the original data onto the selected principal components to obtain the lower-dimensional representation.

**Benefits:**

Dimensionality reduction facilitates easier visualization and interpretation of data. Reduced dimensions often lead to computational efficiency. Principal components capture the most significant patterns in the data.

**More Explanation Regarding SVD Computation:** Consider an $m \times n$ matrix $\boldsymbol{R}$, where $m$ denotes the number of rows, and $n$ represents the number of columns. The primary objective of Singular Value Decomposition (SVD) is to decompose matrix $\boldsymbol{R}$ into three distinct matrices: $\boldsymbol{U}, \boldsymbol{\Sigma}$, and $\boldsymbol{V}^T$ (transpose of matrix $\boldsymbol{V}$). This decomposition is expressed as $\boldsymbol{R} = \boldsymbol{U}\boldsymbol{\Sigma}\boldsymbol{V}^T \in \mathbb{R}^{m \times n}$, as illustrated in Fig. 4.

$\boldsymbol{U}$: An $m \times m$ orthogonal matrix, where its columns signify the left singular vectors of $\boldsymbol{R}$.

$\boldsymbol{\Sigma}$: An $m \times n$ diagonal matrix, featuring singular values of $\boldsymbol{R}$ (non-negative and arranged in descending order).

$\boldsymbol{V}^T$: An $n \times n$ orthogonal matrix, with its columns representing the right singular vectors of $\boldsymbol{R}$.

In addition to singular values and singular vectors, eigenvalues and eigenvectors are also integral to understanding matrix properties. An eigenvalue $\lambda$ and its corresponding eigenvector $\mathbf{v}$ of a square matrix $\boldsymbol{R}$ satisfy the equation $\boldsymbol{R}\mathbf{v} = \lambda\mathbf{v}$. Eigenvectors denote directions in the vector space that are solely scaled by the matrix $\boldsymbol{R}$, while eigenvalues represent the scaling factors for these eigenvectors.

SVD and its Relationship to Eigenvalues and Eigenvectors: SVD establishes a crucial connection between eigenvalues and eigenvectors with the singular values and singular vectors of a matrix. The singular values of $\boldsymbol{R}$ are the square roots of the eigenvalues of either $\boldsymbol{R}\boldsymbol{R}^T$ or $\boldsymbol{R}^T\boldsymbol{R}$, and the left and right singular vectors are the eigenvectors of $\boldsymbol{R}\boldsymbol{R}^T$ and $\boldsymbol{R}^T\boldsymbol{R}$, respectively.

Rank and Matrix Approximation: The rank of a matrix $\boldsymbol{R}$ is determined by the count of non-zero singular values in $\boldsymbol{\Sigma}$. By retaining only the largest singular values and their corresponding singular vectors, it becomes feasible to approximate the original matrix $\boldsymbol{R}$ with a lower-rank approximation. This technique is valuable for tasks such as dimensionality reduction and noise reduction, and we leverage this feature in our approach.

**Properties of SVD:**
The singular values in $\boldsymbol{\Sigma}$ are non-negative and arranged in descending order. The columns of $\boldsymbol{U}$ and $\boldsymbol{V}$ are orthonormal, forming an orthogonal basis for their respective vector spaces. The SVD decomposition is unique, except for the sign of the singular values and the order of the singular vectors. SVD is a potent matrix factorization technique, offering a concise representation of a matrix while preserving essential structural properties. Its applications span diverse fields, including data analysis, image processing, recommendation systems, and more (Deisenroth et al., 2020).

# G PRINCIPAL COMPONENT ANALYSIS FOR COMPUTING MEAN VARIANCE

In this section, we describe the process of calculating the mean variance of high-dimensional features using Principal Component Analysis (PCA). Let $X \in \mathbb{R}^{n \times d}$ be the dataset, where $n$ represents the number of samples, and $d$ is the number of features.

## G.1 FEATURE STANDARDIZATION

PCA is sensitive to the scale of the input data, so the first step is to standardize the features, ensuring each has a mean of zero and a variance of one.

Given the dataset $X = \{X_1, X_2, \ldots, X_n\}$, where each $X_i \in \mathbb{R}^d$ represents a sample with $d$ features, we standardize the data as follows:

$$\mu_j = \frac{1}{n} \sum_{i=1}^{n} X_{ij}, \quad \forall j = 1, 2, \ldots, d, \tag{9}$$

$$X_{\text{centered}} = X - \mu, \tag{10}$$

$$X_{\text{standardized}} = \frac{X_{\text{centered}}}{\sigma}, \quad \sigma_j = \sqrt{\frac{1}{n} \sum_{i=1}^{n} (X_{ij} - \mu_j)^2}. \tag{11}$$

Here, $\mu_j$ is the mean of the $j$-th feature, and $\sigma_j$ is its standard deviation.

## G.2 COVARIANCE MATRIX CALCULATION

After standardizing the data, we compute the covariance matrix $\Sigma \in \mathbb{R}^{d \times d}$, which captures the relationships between features. The covariance matrix is defined as:

$$\Sigma = \frac{1}{n-1} X_{\text{standardized}}^{\top} X_{\text{standardized}}, \tag{12}$$

where $\Sigma_{jk}$ represents the covariance between features $j$ and $k$.

## G.3 EIGENVALUE DECOMPOSITION

We perform an eigenvalue decomposition on the covariance matrix $\Sigma$, which yields the principal components and the amount of variance explained by each. The decomposition is given by:

$$\Sigma = V \Lambda V^{\top}, \tag{13}$$

where $V \in \mathbb{R}^{d \times d}$ is the matrix of eigenvectors (principal components), and $\Lambda = \text{diag}(\lambda_1, \lambda_2, \ldots, \lambda_d) \in \mathbb{R}^{d \times d}$ is a diagonal matrix with the eigenvalues $\lambda_j$, representing the variance explained by the $j$-th principal component.

## G.4 EXPLAINED VARIANCE

The eigenvalues $\lambda_j$ indicate the variance captured by each principal component. The proportion of variance explained by the $j$-th component is calculated as:

$$\text{Explained Variance Ratio} = \frac{\lambda_j}{\sum_{k=1}^{d} \lambda_k}. \tag{14}$$

## G.5 MEAN VARIANCE CALCULATION

The mean variance explained by all principal components is computed by averaging the explained variance ratios:

$$\text{Mean Variance} = \frac{1}{d} \sum_{j=1}^{d} \frac{\lambda_j}{\sum_{k=1}^{d} \lambda_k}. \tag{15}$$

This value represents the average variance explained by each principal component.

