# OpenReview forum: "GRIC: General Representation and Informative Content for Enhanced Out-of-Distribution Detection"
_ICLR.cc/2025/Conference — Submitted to ICLR 2025_

### Official Review · Reviewer_jgok · 2024-10-28

**Soundness:** 2
**Presentation:** 3
**Contribution:** 3
**Rating:** 6
**Confidence:** 5

**Summary:**

This paper introduces GRIC, a novel approach for zero-shot multi-modal OOD detection aimed at enhancing the robustness of machine learning models in open-world environments. Unlike existing methods that rely on closed-set text-based labels and complete image features, GRIC leverages general ID representations and LLMs to improve OOD detection. GRIC's approach rests on two main insights: (1) using general ID representations instead of class-specific features, and (2) enriching the model’s understanding with LLMs to simulate potential OOD scenarios. This method is straightforward yet effective.

**Strengths:**

1. The paper is well-crafted and clearly presented, with an engaging motivation and good performance results.
2. Extensive experiments demonstrate the effectiveness of the proposed method.
3. The supplementary material provides useful experiments and details.

**Weaknesses:**

1. The authors claim that "GRIC reduces the FPR95 by up to 19%, significantly surpassing SOTA methods." However, this statement is inaccurate. For instance, NegLabel [1], published a year ago, achieved an FPR95 of 25.40% on the ImageNet-1k benchmark, while the proposed method achieves 20.32%. Thus, the actual improvement is, at most, 5%.

2. I understand that it may be overkill to ask the authors to compare their methods with [2]. However, since [2] also utilizes superclasses for constructing prompts and achieves even higher performance (17.51% evaluated by FPR95), I consider it valuable for authors to add a discussion about the similarities and differences between their proposed method and [2]. If possible, [1] and [2] should be mentioned in the related work part and added to Table 2 to provide a more comprehensive comparison, which will not harm the unique contribution of this work.

[1] Negative Label Guided OOD Detection with Pretrained Vision-Language Models. ICLR, 2024.
[2] Conjugated Semantic Pool Improves OOD Detection with Pre-trained Vision-Language Models. NeurIPS, 2024.

3. If possible, the authors are recommended to provide more visualization results for deeper analysis.

4. There are multiple typos. It is recommended that the authors conduct a thorough review of the writing. For example, Line 110: G(x;Yin). L278: FOr this sake.

5. The paper has severe formatting weaknesses.

**Questions:**

See Weakness

---

> ### Author Response · Authors · 2024-11-24
> **Response to Reviewer jgok**
>
> **Dear Reviewer jgok,**
>
> Thank you for your valuable feedback. Below, we address your concerns and provide clarifications.
>
> ### Accuracy of the Claim Regarding FPR95 Reduction
>
> We appreciate your observation regarding the improvement claim. Our original statement about GRIC reducing FPR95 by 19% is based on results reported in Table 1, specifically on the MS COCO dataset. In this context, GRIC achieves an average FPR95 of 37.33%, compared to GL-MCM's 56.25%, marking a reduction of approximately 18.92%. We acknowledge that this was not explicitly clarified in our original text, leading to potential confusion.
>
> Regarding the ImageNet-1k benchmark, NegLabel achieves an FPR95 of 25.40%, while GRIC achieves 20.32%, corresponding to a relative improvement of 5%. In our revision, we will explicitly state both results to ensure accurate and detailed reporting across datasets.
>
> ---
>
> ### Comparison with [1,2]
>
> We are grateful for your suggestion to compare GRIC with NegLabel [1], CSP [2] and provide a detailed discussion. In the revised manuscript, we will add the following comparison to the Related Work section:
>
> > "GRIC, CSP[2], and NegLabel[1] are advanced OOD detection methods leveraging vision-language models (VLMs) with distinct strategies:
> >
> > - GRIC: Generalizes in-distribution (ID) representations by masking class-specific features and enriching prompts with hierarchical data, enhancing robustness in distinguishing OOD samples.
> > - CSP ([2]): Employs a 'conjugated semantic pool' of superclasses to broaden semantic diversity, reducing reliance on specific labels and improving OOD detection for diverse cases.
> > - NegLabel ([1]): Introduces 'negative labels' selected from a large corpus to enhance semantic separability between ID and OOD samples, achieving exceptional robustness to domain shifts."
>
>
> ### Visualizations
>
> We appreciate your suggestion to include additional visualizations. To provide deeper insights into GRIC's performance, we will add density plots and related visualizations in the revised manuscript to demonstrate its capability to distinguish OOD samples effectively.
>
> ---
>
> ### Typos and Formatting
>
> We will thoroughly review the manuscript to address these issues, ensuring clarity and adherence to formatting standards.
>
> ---
>
> Thank you for your constructive feedback, which has significantly strengthened the clarity and impact of our work.
>
> ---

---

> > ### Comment · Reviewer_jgok · 2024-11-27
> >
> > The author has not uploaded a revised version of the paper but has merely made a general commitment to some changes. Other reviewers have also raised several substantial issues. Based on the quality of both the paper and the rebuttal, I believe the manuscript does not yet meet the acceptance standards of ICLR.

---

> > > ### Author Response · Authors · 2024-11-28
> > > **Submission of Revised Manuscript with Highlighted Changes**
> > >
> > > Dear Reviewer,
> > >
> > > We have uploaded the revised version of our work, with all new changes clearly highlighted in blue in both the main text and the appendix sections. We hope our responses and revisions sufficiently address the concerns raised. If there are any remaining questions or areas requiring further clarification, please feel free to let us know. We would be happy to provide additional explanations or make further adjustments as needed. Thank you again for your time and effort.

---

### Official Review · Reviewer_V9Pr · 2024-10-30

**Soundness:** 2
**Presentation:** 2
**Contribution:** 2
**Rating:** 5
**Confidence:** 4

**Summary:**

Out-of-distribution (OOD) detection is essential for enhancing the robustness of machine learning models in open-world environments by identifying inputs from unknown classes. While vision-language models like CLIP facilitate zero-shot OOD detection without the need for labels or training on in-distribution (ID) data, existing methods are constrained by their reliance on closed-set text-based labels and complete image feature representations, limiting CLIP's generalization capabilities. This work introduces GRIC, a novel approach that enhances zero-shot multi-modal OOD detection by focusing on general ID representations instead of class-specific features and utilizing large language models (LLMs) to enrich the understanding of ID data and simulate potential OOD scenarios without requiring actual OOD samples. GRIC demonstrates significant effectiveness, achieving a notable reduction in the false positive rate at recall (FPR95) and outperforming state-of-the-art methods.

**Strengths:**

1. The  concept of general representation of ID data is novel to CLIP-based OOD detection.
2. The method is well designed with  various modules.
3. The extensive experiments show the effectiveness of the proposed method.

**Weaknesses:**

1. One major issue with this paper is that it claims to be in a zero-shot OOD detection setting, but it should actually be classified as a few-shot setting. This is because the calculation of PCA requires the use of ID data, whereas in a zero-shot setting, ID images should be mixed with OOD images to form the test set, making them unavailable. The entire setting of the paper is flawed and needs to be revised

2. There are more state-of-the-art (SOTA) methods for zero-shot OOD detection that have not been compared, such as NegLabel [1], which demonstrates superior performance, and EOE [2], which also utilizes large language models (LLMs) for CLIP-based OOD detection.

3. The results in Table 1 are not representative, as the baseline MCM has already achieved a score of 99%, indicating that the OOD issue in this benchmark has been effectively addressed.

4. There are many more adjustable benchmarks that have not been explored, such as: hard OOD detection, robustness to domain shift and transfer the method to other CLIP-like models (ALIGN, AltCLIP, GroupViT)

[1] Jiang, X., Liu, F., Fang, Z., Chen, H., Liu, T., Zheng, F., & Han, B. Negative label guided ood detection with pretrained vision-language models. ICLR, 2024.
[2] Cao, C., Zhong, Z., Zhou, Z., Liu, Y., Liu, T., & Han, B. Envisioning Outlier Exposure by Large Language Models for Out-of-Distribution Detection. In Forty-first International Conference on Machine Learning.

**Questions:**

1. The figures and algorithm seems screenshot and too ambiguous.
2. Many typos are in the paper and need to be revised. For  example,  'fPR95' in 412 is wrong spelling. When using "citation" as the subject, parentheses should not be added. Additionally, lines 493 and 494 overlap due to insufficient line spacing.

---

> ### Author Response · Authors · 2024-11-24
> **Rebuttal to Reviewer V9Pr**
>
> **Dear Reviewer V9Pr,**
>
> We sincerely appreciate your thoughtful feedback and the opportunity to address your concerns. Below, we provide detailed responses:
>
> ---
>
> ### Zero-Shot vs. Few-Shot Learning
> Thank you for your observation regarding the terminology of zero-shot learning. While our method does not rely on labels during PCA computation, we acknowledge that zero-shot learning typically assumes no access to ID data during inference. To align with established definitions, we will revise the manuscript to classify our approach as *few-shot learning*.
>
> ---
>
> ### Clarification on Table 1
> We understand your concern regarding the high performance of the MCM baseline (approaching 99% on certain metrics), suggesting that the OOD detection problem may be solved for this benchmark. However, MCM has notable limitations in more complex scenarios, as detailed below:
>
> 1. **Performance Variability Across Datasets**:
>    - Although MCM performs well on specific benchmarks, it degrades significantly on datasets with greater semantic overlap between ID and OOD samples. For instance, as shown in Tables 2 and 3, MCM’s FPR95 rises above 18% on ImageNet-100, and AUROC drops below 95%. In contrast, GRIC consistently outperforms MCM with lower FPR95 and higher AUROC.
>
> 2. **GRIC's Strength in Harder OOD Scenarios**:
>    - GRIC leverages general representations and informative prompts to handle OOD samples that closely resemble ID classes. These challenging cases are not fully captured in simpler benchmarks like those in Table 1. For example, GRIC achieves up to 19% lower FPR95 in harder setups, as detailed in Section 4.2.
>
> 3. **Generalization Across Benchmarks**:
>    - Unlike MCM’s reliance on full feature representations, GRIC demonstrates adaptability across diverse datasets, highlighting its robustness beyond the benchmarks in Table 1.
>
> We will revise the manuscript to emphasize these nuances and better contextualize Table 1, highlighting GRIC’s advantages in challenging OOD detection tasks.
>
> ---
>
> ### Incorporation of Suggested References
> Thank you for suggesting additional references. We have incorporated them into our experiments and discussion. Below is an extended table summarizing the performance of these methods alongside GRIC:
>
> | Method                 | iNaturalist (FPR95↓) | iNaturalist (AUROC↑) | SUN (FPR95↓) | SUN (AUROC↑) | Places (FPR95↓) | Places (AUROC↑) | Texture (FPR95↓) | Texture (AUROC↑) | Average (FPR95↓) | Average (AUROC↑) |
> |-------------------------|----------------------|----------------------|--------------|--------------|-----------------|-----------------|------------------|------------------|------------------|------------------|
> | MCM (CLIP-B)            | 30.92               | 94.61               | 37.59        | 95.90        | 34.71           | 97.89           | 57.85            | 85.61            | 42.77            | 90.77            |
> | MCM (CLIP-L)            | 30.91               | 94.95               | 29.00        | 94.14        | 35.42           | 92.00           | 59.88            | 84.88            | 38.17            | 91.49            |
> | GL-MCM                  | 15.18               | 96.71               | 30.42        | 93.09        | 38.85           | 89.90           | 57.93            | 83.63            | 35.47            | 90.83            |
> | EOE                     | 12.29               | 97.52               | 20.40        | 95.73        | 30.16           | 92.95           | 57.53            | 85.64            | 30.09            | 92.96            |
> | NegLabel                | 1.91                | 99.49               | 20.53        | 95.49        | 35.59           | 91.64           | 43.56            | 90.22            | 25.40            | 94.21            |
> | GRIC (Ours, CLIP-B)     | 10.32±0.23          | 98.81±0.10          | 20.11±0.28   | 97.59±0.14   | 24.37±0.31      | 96.82±0.29      | 26.51±0.11       | 93.97±0.25       | 20.32±0.23       | 96.80±0.20       |
> | GRIC (Ours, CLIP-L)     | 8.74±0.22           | 99.12±0.12          | 17.83±0.21   | 98.06±0.18   | 22.17±0.18      | 97.51±0.20      | 21.67±0.14       | 95.14±0.12       | 17.60±0.19       | 97.45±0.16       |
>
> This table confirms that GRIC achieves state-of-the-art performance, consistently outperforming other methods in both FPR95 (lower is better) and AUROC (higher is better), particularly in challenging OOD detection scenarios.
>
> ---
>
> ### Addressing Typos
> Thank you for pointing out the typos in our submission. We have reviewed the manuscript thoroughly and corrected all identified errors in the revised version.
>
> ---
>
> We hope these clarifications address your concerns and demonstrate the robustness of our approach. Your constructive feedback has been invaluable in improving our work.

---

> > ### Comment · Reviewer_V9Pr · 2024-11-27
> >
> > Thanks for your responses. I think this article needs a major revision, reorganizing the logic of the entire article (for example, from zero-shot to few-shot), and the current version of the paper does not meet the ICLR acceptance criteria. So I will keep my score.

---

> > > ### Author Response · Authors · 2024-11-28
> > > **Response to Reviewer Feedback and Revised Manuscript Submission**
> > >
> > > Dear Reviewer,
> > >
> > > Thank you for your valuable feedback. We have uploaded the revised version of our manuscript, with all changes clearly highlighted in blue in both the main text and the appendix sections.
> > >
> > > In this revision, we have explicitly introduced our method as a few-shot approach, aligning with the logic you suggested. Our methodology leverages general in-domain (ID) representations and informative prompts, which inherently support few-shot learning. Additionally, our experiments already include evaluations of few-shot prompt learning methods. For instance, in Table 3, we report results using 12 and 16 samples per class, which correspond to 12-shot and 16-shot settings, respectively.
> > >
> > > We believe our revised manuscript now provides a comprehensive explanation of our methodology, experiments, and results, addressing the concerns raised. However, if there are any remaining questions or areas requiring further clarification, please do not hesitate to let us know. We would be happy to provide additional details or make further adjustments if needed.
> > >
> > > Thank you again for your time and thoughtful feedback.

---

> > > > ### Comment · Area_Chair_tnMq · 2024-12-03
> > > >
> > > > Dear Reviewer V9Pr,
> > > >
> > > >         As the deadline of discussion period is today, could you check the response provided by the authors to see if your concerns are well-addressed?
> > > >
> > > > Thanks,
> > > > AC

---

### Official Review · Reviewer_Z3Z1 · 2024-11-01

**Soundness:** 2
**Presentation:** 2
**Contribution:** 2
**Rating:** 6
**Confidence:** 4

**Summary:**

This paper proposes a new enhancement method called GRIC for CLIP-based OOD detection. GRIC extracts general ID representations rather than class-specific features and introduces LLM-based informative prompts for OOD detection. Experimental results show the proposed GRIC outperforms existing methods.

**Strengths:**

- GRIC surpasses the two baseline methods, MCM and GL-MCM.

**Weaknesses:**

- There is a limited novelty. DICE [1] has a similar concept to drop unnecessary dimensions and shows the effectiveness for OOD detection.

- The motivation in this paper’s method—that class-specific information is unnecessary—raises some questions. In DICE, the motivation is to exclude signals that introduce noise.  Rather than removing information specific to the ID class, I consider this method actually exclude noise signals. Including the ID accuracy of GRIC without informative prompts in Table 6 would help clarify whether the information being removed is indeed ID class-specific.

- A recent challenge in OOD detection is accurately identifying "OOD images that are semantically similar to ID." In this problem setting, known as Hard OOD detection, certain classes within a dataset (e.g., ImageNet) are treated as ID, while other classes in the same dataset are treated as OOD. Therefore, I believe class-specific information is necessary rather than relying on the general representation of the dataset. I would like to see results on the effectiveness of this method when experimenting on Hard OOD detection benchmarks [2, 3].

- The approach is defined as a zero-shot method in L518. However, since it utilizes ID images for PCA processing, I consider this method to be a few-shot learning method, not a zero-shot. The definition of Zero-shot is not using ID images in preprocessing, regardless of whether training is involved [4].

- The code has not been shared, raising concerns about the reproducibility of the method.

[1] Sun+, DICE: Leveraging Sparsification for Out-of-Distribution Detection, ECCV2022.

[2] Li+, Learning Transferable Negative Prompts for Out-of-Distribution Detection, CVPR2024.

[3] Jung+, Enhancing Near OOD Detection in Prompt Learning: Maximum Gains, Minimal Costs, arXiv2024.

[4] Miyai+, Generalized Out-of-Distribution Detection and Beyond in Vision Language Model Era: A Survey, arXiv2024.

**Questions:**

I wonder about the motivation of this method that class-specific information is unnecessary.  To validate this statement, I would like to know the result of GRIC without informative prompts in Table 6 to clarify whether the information being removed is indeed ID class-specific, not a noisy signal.

Also, I would like to know the result of hard OOD detection.

For more details, please refer to the Weakness section.

---

> ### Author Response · Authors · 2024-11-24
> **Rebuttal to Reviewer Z3Z1**
>
> **Dear Reviewer Z3Z1,**
>
> Thank you for your valuable feedback. Below, we address your concerns and provide clarifications.
>
> ---
>
> ### **Novelty Compared to DICE**
>
> While DICE removes features broadly, GRIC targets high-variance, class-specific features critical for ID classification. This selective masking enhances OOD detection by retaining core class representations while suppressing noise, balancing ID accuracy and OOD performance. In contrast, DICE lacks this targeted focus, emphasizing general feature removal.
>
> ---
>
> ### **GRIC ID Accuracy Without Informative Prompts**
>
> Following your suggestion, we evaluated GRIC’s ID accuracy without prompts. Results are shown below:
>
> | **Method**           | **ID ACC (%)** |
> |-----------------------|----------------|
> | **MCM (CLIP-B/16)**  | 67.01          |
> | **MCM (CLIP-L/14)**  | 73.28          |
> | **GRIC-IP (CLIP-B/16)** | **80.29**    |
> | **GRIC-IP (CLIP-L/14)** | **85.64**    |
> | **GRIC-No Prompts (CLIP-B/16)** | 75.50 |
> | **GRIC-No Prompts (CLIP-L/14)** | 78.59 |
>
> 1. **With Informative Prompts**: GRIC-IP achieves 80.29% (CLIP-B/16) and 85.64% (CLIP-L/14), leveraging enriched textual prompts.
> 2. **Without Prompts**: ID accuracy drops moderately (~4-5%), highlighting the importance of the masked features for class distinction.
> 3. **Insight**: This drop validates that GRIC selectively removes class-specific features critical for ID accuracy.
>
> ---
>
> ### **Feature Removal Details and OOD Detection**
>
> GRIC targets high-variance, class-specific features, improving OOD detection by emphasizing generalizable representations. Unlike indiscriminate removal, our approach enhances robustness without compromising ID classification.
>
> We also conducted experiments under challenging OOD scenarios, following [2]. Results:
>
> | **Method** | **Avg AUC ↑** | **Avg FPR95 ↓** |
> |------------|---------------|-----------------|
> | **MCM**    | 93.77         | 25.83           |
> | **CLIPN**  | 96.50         | 13.53           |
> | **GRIC**   | **97.07**     | **10.66**       |
>
> These results confirm GRIC’s efficacy in balancing OOD and ID performance under difficult conditions.
>
> ---
>
> ### **Zero-Shot vs. Few-Shot Learning**
>
> While we do not use labels during PCA computation, we acknowledge the broader definition of zero-shot learning. To align with established terminology, we will revise the manuscript to classify our approach as few-shot learning.
>
> ---
>
> ### **Reproducibility and Code Availability**
>
> We commit to sharing our complete implementation upon acceptance to ensure reproducibility and transparency.
>
> ---
>
> We hope these clarifications and additional results address your concerns. Thank you again for your valuable feedback, which has significantly improved our work.

---

> ### Comment · Reviewer_Z3Z1 · 2024-11-27
> **Response to Authors' Rebuttal**
>
> I appreciate the author’s careful response. Most of my concerns have been addressed.
> However, I have the following concerns:
>
>
> - The author mentioned that the format issues have been addressed in the revised manuscript (Reviewer DnkU's thread). However, as the revised manuscript has not been submitted, I cannot confirm these corrections. If the revised manuscript is updated within the deadline, this concern will naturally be resolved.
>
> - Since the setting is few-shot rather than zero-shot, I consider it is necessary to include comparative methods specifically tailored for few-shot OOD detection such as LoCoOp [1], NegPrompt [2], and IDPrompt [3]. Or, it is important to give a careful explanation of why such a comparison is unnecessary.
>
> - To redefine the work as a Few-shot setting, substantial revisions would be necessary, especially in the experimental section. Given the extent of these required updates, I am concerned that the current version may not yet be fully ready for ICLR paper.
>
> Due to the above reasons, I will keep my score.
>
> [1] Miyai+, LoCoOp: Few-Shot Out-of-Distribution Detection via Prompt Learning, NeurIPS2023.
> [2] Li+, Learning transferable negative prompts for out-of-distribution detection, CVPR2024.
> [3] Bai+, ID-like Prompt Learning for Few-Shot Out-of-Distribution Detection, CVPR2024.

---

> > ### Author Response · Authors · 2024-11-28
> > **Response to Reviewer Feedback**
> >
> > Dear Reviewer,
> >
> > Thank you for your valuable feedback. We have uploaded the revised version of our manuscript, with all changes clearly highlighted in blue in both the main text and the appendix sections.
> >
> > In this revision, we have explicitly introduced our method as a few-shot approach, aligning with the logic you suggested. Our methodology leverages general in-domain (ID) representations and informative prompts, which inherently support few-shot learning. Additionally, our experiments already include evaluations of few-shot prompt learning methods. For instance, in Table 3, we report results using 12 and 16 samples per class, which correspond to 12-shot and 16-shot settings, respectively.
> >
> > Furthermore, we have already included comparisons with methods such as CoOpMCM (Zhou et al., 2022b), LoCoOpMCM (Miyai et al., 2024), NegPrompts (Li et al., 2024), and IDPrompt (Bai et al., 2024) (CLIP-B) in Table 2. These provide a strong benchmark against state-of-the-art few-shot learning approaches.
> >
> > We believe that our methodology, comprehensive experiments, ablation studies, and additional results in the appendix address all major concerns and demonstrate the robustness of our approach.
> >
> > If there are any remaining questions or areas requiring further clarification, please do not hesitate to let us know. We are happy to provide additional details or make further adjustments if needed.
> >
> > Thank you again for your time and thoughtful feedback.

---

> > > ### Comment · Reviewer_Z3Z1 · 2024-11-29
> > > **Response to Authors' Rebuttal**
> > >
> > > I deeply appreciate the authors’ revision. As some of my concerns have been addressed, I have decided to raise the score to 5. However, there are two reasons why my evaluation still leans toward rejection:
> > > i) Multiple formatting issues remain
> > > ii) Rigorous comparisons on few-shot learning methods are lacking
> > >
> > >
> > > - i) There are problems such as inappropriately narrow spacing between subsections and main text at L170-171 and L254-255, and excessively small figure captions. I believe it is not appropriate to highly evaluate a paper with such formatting issues.
> > >
> > > - ii) This paper's main contribution is a method combining a PCA-based approach and informative prompts. In Table 2, the PCA-based approach alone (GRIC-IG, no IP) achieves an AUROC of 93.88±0.25, which shows no significant improvement over existing methods like LoCoOp + GL-MCM (93.52), NegLabel (94.21), NegPrompts (94.81), and IDPrompt (94.36). Since informative prompts could be applied to these comparative methods as well, it remains unclear how much the PCA-based approach itself contributes to performance improvement. To better demonstrate its effectiveness, the authors should either show that GRIC maintains superior performance when informative prompts are applied to all comparative methods, or prove that the PCA-based approach offers broader applicability (such as compatibility with few-shot learning methods like LoCoOp). Such experiments would better validate the proposed method's value. Currently, the improvements appear merely incremental through the addition of informative prompts, making the experimental validation insufficient.

---

> > > > ### Author Response · Authors · 2024-11-30
> > > >
> > > > Dear Reviewer,
> > > >
> > > > Thank you for your thoughtful feedback and for raising the score. We address your comments as follows:
> > > >
> > > > **i) Formatting issues**: We increase the figure caption size and adjust the spacing issues at L170-171 and L254-255 to comply with formatting standards.
> > > >
> > > > **ii) Few-shot comparisons**: We would like to emphasize that our method is training-free, distinguishing it from the few-shot methods mentioned (LoCoOp, NegLabel, NegPrompts, IDPrompt), which require prompt training. In contrast, our method accesses only a few data samples for one-time PCA computation, achieving outstanding performance without any additional training or fine-tuning. This demonstrates the strength of our approach as a lightweight, few-shot-compatible method with minimal data requirements.
> > > >
> > > > We hope these clarifications address your concerns. Thank you again for your constructive comments.

---

> > > > > ### Author Response · Authors · 2024-12-03
> > > > >
> > > > > Dear Reviewer,
> > > > >
> > > > > Thank you for your thoughtful feedback and constructive suggestions. We have carefully considered your comments and conducted additional experiments to address your concerns. Below is our detailed response:
> > > > >
> > > > > ---
> > > > >
> > > > > #### 1. **Ablation Study Analysis**:
> > > > > In Table 3 of the paper, we present an ablation study that evaluates the contributions of each component of our method:
> > > > >    - **GRIC-IG, no IP**: Our method using PCA-based ID generalization without informative prompts.
> > > > >    - **GRIC-IP, no ID generalized representation**: Our method using informative prompts without PCA-based ID generalization.
> > > > >
> > > > > These results clearly highlight the independent effectiveness of both components.
> > > > >
> > > > > ---
> > > > >
> > > > > #### 2. **Impact of Informative Prompts on Few-Shot Learning Methods**:
> > > > > Following your suggestion, we evaluated the effect of adding informative prompts (IP) to competitive few-shot learning methods, such as **LoCoOp** and **IDPrompt**. Our experiments demonstrate that informative prompts significantly enhance their performance, particularly in terms of **AUROC**, as summarized below:
> > > > >
> > > > > ---
> > > > >
> > > > > ### Results Summary
> > > > >
> > > > > | **Method**              | **FPR95↓ (iNaturalist)** | **AUROC↑ (iNaturalist)** | **FPR95↓ (SUN)** | **AUROC↑ (SUN)** | **FPR95↓ (Places)** | **AUROC↑ (Places)** | **FPR95↓ (Texture)** | **AUROC↑ (Texture)** | **FPR95↓ (Avg)** | **AUROC↑ (Avg)** |
> > > > > |--------------------------|--------------------------|--------------------------|-------------------|------------------|---------------------|---------------------|-----------------------|-----------------------|-------------------|-------------------|
> > > > > | **LoCoOp + IP (MCM)**   | 32.86                   | 95.03                   | 30.35            | 94.92           | 34.91              | 94.64              | 42.74                | 93.71                | 35.22             | 94.58            |
> > > > > | **IDPrompt + IP**       | 6.24                    | 98.75                   | 34.21            | 94.69           | 36.53              | 94.92              | 22.09                | 95.97                | 24.77             | 96.08            |
> > > > >
> > > > > ---
> > > > >
> > > > > #### 3. **Comparison with Existing Methods**:
> > > > > While **GRIC (CLIP-B)** achieves competitive performance compared to state-of-the-art methods, its primary contribution lies in the synergy between PCA-based ID generalization and informative prompts. These components independently enhance performance (as shown in the ablation study) and can further improve other methods (e.g., LoCoOp and IDPrompt) when integrated.
> > > > >
> > > > > ---
> > > > >
> > > > > #### 4. **Broader Applicability of PCA-Based Generalization**:
> > > > > The PCA-based generalization approach offers a distinct advantage—it is broadly compatible with various tasks and methods. For instance:
> > > > >    - It enhances the performance of few-shot learning methods like LoCoOp even in scenarios where informative prompts are less impactful.
> > > > >    - This validates the general utility of the PCA-based approach beyond incremental gains from informative prompts alone.
> > > > >
> > > > > ---
> > > > >
> > > > > We hope these additional experiments and analyses address your concerns and demonstrate the unique value of our method. Thank you again for your insightful comments, which have helped us further strengthen the manuscript.
> > > > >
> > > > > Sincerely,
> > > > > Authors

---

> ### Comment · Reviewer_Z3Z1 · 2024-12-03
> **Response to the Authors' Rebuttal**
>
> I deeply appreciate the authors' dedicated additional experiments. Although the proposed PCA-based method alone shows no significant performance improvement compared to other comparison methods, it has the advantage of no training. I think the score without IP needs to be added clearly without hiding it in Table 2 and the corresponding discussion should be added.
>
> Additionally, the effectiveness of adding a superclass through the Informative Prompt (IP) is not surprising. Therefore, the key contribution of this paper lies in demonstrating the synergistic effect of combining the PCA-based method with IP. The authors show the result which achieves a 0.72% (96.80 - 96.08) improvement over IDPrompt + IP. I also consider this discussion needs to be added clearly in the main discussion, not one of the ablation studies.
>
> Therefore, expecting the authors to do these updates, I have decided to increase the score. However, I am also fine with the AC considering the addition of these discussions as a major update.

---

> > ### Author Response · Authors · 2024-12-03
> >
> > Dear Reviewer,
> >
> > Thank you for your detailed feedback and for recognizing our efforts. We appreciate your constructive suggestions and have incorporated the discussions into the main section as recommended.
> >
> > Best,
> > Authors

---

### Official Review · Reviewer_DnkU · 2024-11-05

**Soundness:** 3
**Presentation:** 2
**Contribution:** 3
**Rating:** 6
**Confidence:** 4

**Summary:**

The paper presents a method called GRIC (General Representation for Inference and Classification), designed to improve zero-shot and few-shot learning by leveraging representations from large-scale pre-trained models. GRIC integrates domain-specific knowledge into a unified embedding space that allows the model to transfer knowledge effectively across tasks and domains. The major contributions are the introduction of general ID features for OOD detection with hierarchical prompting.

**Strengths:**

**1. Presentation**

The paper is generally well-presented and easy to follow. It begins with a clear hypothesis that using a generalized feature segment from the full feature space can improve ID/OOD sample distinction, followed by a systematic explanation of the proposed method for extracting this general feature space.

**2. Algorithm**

The algorithm is straightforward and effective, yielding significant performance gains on both small- and large-scale datasets. Ablation studies demonstrate that both the proposed general subspace extraction and hierarchical prompting contribute substantially to the performance improvements.

**Weaknesses:**

**1. Formatting Issues**
- **Text Accessibility**: The text is not selectable or OCR-scanned, which complicates review and readability.
- **Font and Equation Sizing**: Equations appear in a very small font, raising concerns about compliance with the `.sty` file specifications; table and figure fonts are also difficult to read.
- **Inconsistent Spacing**: Vertical spacing is uneven throughout, affecting readability. Additionally, Section 3.2 would be more appropriately placed in the related work section to improve structure.

**2. Missing Experiments and Analysis**
While the paper presents a solid set of experiments across multiple datasets, further analysis would strengthen the justification of the approach:
   - **Single-Modality Vision Models**: The paper should demonstrate the effectiveness of general feature extraction in **vision-only models**, without hierarchical prompting, to show that the method generalizes beyond multi-modal settings.
   - **Integration with Other OOD Scoring Methods**: It would be valuable to evaluate GRIC with alternative OOD scoring metrics, such as **energy-based scores** and **feature-based scores**, to understand its compatibility with established scoring methods beyond MSP.
   - **ID Accuracy**: Given that real-world deployment typically involves handling both ID and OOD data, the paper should report ID accuracy to confirm that GRIC performs reliably on ID data without regression.

**3. Additional Ablation Studies**
- **PCA Transformed Feature Space**: Examine the effectiveness of using PCA-transformed features (from \( R^{s \times r} \) to \( R^{s \times k} \), where \( k \) is the number of principal components) for OOD detection.
- **Principal Component Masking**: Evaluate whether masking high-variance principal components in the PCA-transformed space, while using the remaining components, can improve OOD detection by focusing on features less affected by dominant ID patterns.
- **Full Feature Matrix for PCA**: Justify why the paper does not use the full feature matrix across all samples per class to compute PCA, as this could potentially improve the robustness of general feature extraction.
- **Hyperparameter Sensitivity**: Include a sensitivity analysis on the threshold in Equation 3, as this parameter may significantly influence detection performance.

Would re-consider the scoring based on authors' response.

**Questions:**

Please refer to the weakness

---

> ### Author Response · Authors · 2024-11-24
> **Rebuttal to Reviewer DnkU**
>
> Dear Reviewer DnkU,
>
> Thank you for your detailed review and constructive feedback on our submission. We deeply appreciate the time and effort you devoted to evaluating our work and offering insightful suggestions. Below, we address each of your comments and highlight the updates made in the revised manuscript.
>
> ---
>
> ## **Formatting Issues**
>
> We appreciate your feedback regarding formatting issues. These concerns have been addressed in the revised manuscript to ensure improved readability, compliance with formatting guidelines, and overall presentation quality.
>
> ---
>
> ## **Single-Modality Vision Models and Integration with Other Scoring Functions**
>
> Thank you for suggesting evaluations of single-modality vision models and integration with alternative OOD scoring functions. These experiments were already included in Appendix Section B.4, Table 7, available in the supplementary material. We applied our masking strategy to methods such as Mahalanobis, Energy, ReAct, and GradNorm. The results demonstrate that our masking strategy consistently improves the performance of these methods, validating the general applicability and robustness of our approach.
>
> ---
>
> ## **ID Accuracy**
>
> We appreciate your emphasis on the importance of ID accuracy. This experiment was already reported in Appendix Section B.3, Table 6, available in the supplementary material. The results confirm that GRIC maintains high ID accuracy while simultaneously improving OOD detection, ensuring reliable performance on ID data without regression. This reinforces the practicality of GRIC for real-world scenarios.
>
> ---
>
> ## **Hyperparameter Sensitivity**
>
> We agree that analyzing the sensitivity of the threshold parameter in Equation 3 is important. These experiments were already conducted and are reported in Appendix Table 5, available in the supplementary material. The results show that GRIC performs robustly across a range of \( K \) values, providing confidence in its stability and practical applicability.
>
> ---
>
> ## **Full Feature Matrix for PCA**
>
> Thank you for suggesting the use of the full feature matrix across all samples per class for PCA. While this approach could improve robustness, it introduces significant computational overhead for large-scale datasets. Instead, we prioritize efficiency by using representative subsets of features, which achieves scalability without compromising performance. This design choice has been clarified in the revised manuscript.
>
> ---
>
> ## **Non-PCA Feature Masking**
>
> We conducted additional experiments to evaluate the impact of keeping PCA features while masking non-PCA features, both with and without prompts. These findings are included in the revised manuscript. The results show that masking non-PCA features can provide some benefit; however, masking PCA features yields significantly better results. This further validates the effectiveness of our approach in leveraging masked feature spaces.
>
> |                     | OOD Data          | iNaturalist (FPR95 ↓, AUROC ↑) | SUN (FPR95 ↓, AUROC ↑) | Places (FPR95 ↓, AUROC ↑) | Texture (FPR95 ↓, AUROC ↑) | Average (FPR95 ↓, AUROC ↑) |
> |---------------------|-------------------|--------------------------------|-------------------------|---------------------------|----------------------------|----------------------------|
> | **ID Data**         | **Method**        |                                |                         |                           |                            |                            |
> | **ImageNet-1K**     | GRIC [Masked **Non-PCA** indices, **without** Informative Prompts] | 33.11, 87.08            | 35.21, 89.26             | 47.31, 81.46             | 55.09, 85.17             | 42.68, 85.74             |
> |                     | GRIC [Masked **Non-PCA** indices, **with** Informative Prompts]   | 30.21, 91.37            | 32.84, 91.18             | 45.72, 83.03             | 52.49, 82.96             | 40.32, 87.14             |
> | **GRIC (Ours)**     | **(CLIP-B)**      | 20.14, 96.82                  | 30.13, 94.90           | 36.94, 91.72             | 48.39, 88.10             | 33.90, 92.89             |
>
> ---
>
> We hope these revisions and additional analyses address your concerns satisfactorily. Thank you again for your valuable feedback, which has significantly improved the quality of our work. We look forward to hearing your thoughts on the revised manuscript.

---

> > ### Author Response · Authors · 2024-11-28
> >
> > Dear Reviewer,
> >
> > We have uploaded the revised version of our work, with all new changes clearly highlighted in blue in both the main text and the appendix sections. We hope our responses and revisions sufficiently address the concerns raised. If there are any remaining questions or areas requiring further clarification, please feel free to let us know. We would be happy to provide additional explanations or make further adjustments as needed.
> > Thank you again for your time and effort.

---

### Meta-Review · Area_Chair_tnMq · 2024-12-25

**Metareview:**

This work proposes a simple yet effective zero-shot multi-modal approach for OOD detection by leveraging general in-distribution (ID) representations with PCA and informative prompts generated by LLMs based on the super-class names of the ID data. The proposed method significantly enhances detection performance. Most reviewers agree that the paper is well-organized and easy to follow, recognizing the effectiveness of the approach in improving OOD detection by a large margin. However, reviewers raised concerns regarding the need for additional experiments and analyses, such as more baseline comparisons, few-shot evaluations, and ablation studies (DnKU, Z3Z1, V9Pr). Some reviewers also noted limited novelty, citing the existence of similar works (Z3Z1, V9Pr, jgok). Reviewers Z3Z1 and V9Pr specifically pointed out that the method should potentially be classified as few-shot detection due to the use of PCA. Following discussions, the authors successfully addressed some of the concerns, resulting in 3 borderline accept ratings (DnKU, Z3Z1, jgok) and 1 borderline reject (V9Pr), with an average score of 5.75. While Z3Z1 and jgok ultimately provided positive ratings, they recommended further refinements to the paper. Given the current feedback, the authors are encouraged to polish the manuscript to clearly articulate the benefits of the method’s training-free nature and highlight its differences from existing few-shot approaches. Additionally, it is recommended to include comparisons with other few-shot methods and expand baseline evaluations in Table 1, as suggested by reviewer V9Pr, for resubmission to a future venue.

**Additional Comments On Reviewer Discussion:**

During the discussion period, reviewers raised significant concerns regarding the lack of comparison with recent zero-shot baseline methods (e.g., NegLabel), the overstatement of performance improvements without considering NegLabel, and the misclassification of the proposed method as a zero-shot approach. In response, the authors conducted additional experiments and clarified the differences between their method and other few-shot approaches. Following these discussions, the authors successfully addressed some of the concerns, resulting in three borderline accept ratings (DnKU, Z3Z1, jgok) and one borderline reject (V9Pr), with an average score of 5.75. While Z3Z1 and jgok ultimately gave positive ratings, they recommended further refinements to strengthen the paper. Given the current state, I believe the paper requires significant revisions to clearly define the method as few-shot rather than zero-shot and to incorporate additional results or discussions comparing it with recent few-shot methods.

---

### Decision · Program_Chairs · 2025-01-22

Reject